# IR-OptSet: An Optimization-Sensitive Dataset for Advancing LLM-Based IR Optimizer

**Zi Yang**[1,2]*, **Lei Qiu**[1,3]*, **Fang Lyu**[1]†, **Ming Zhong**[1], **Zhilei Chai**[2]
**Haojie Zhou**[2], **Huimin Cui**[1,3]†, **Xiaobing Feng**[1,3]

[1] SKLP, ICT, CAS, China    [2] Jiangnan University, China    [3] UCAS, China

{flv, cuihm}@ict.ac.cn

## Abstract

Compiler optimization is essential for improving program performance, yet modern compilers still depend on manually crafted transformation rules over intermediate representations (IRs). As compilers grow in complexity, maintaining these rule-based optimizations becomes increasingly labor-intensive and difficult to scale. Recent advances in large language models (LLMs) offer a promising alternative, but their effectiveness in compiler optimization remains limited – primarily due to the lack of IR-oriented datasets that expose models to diverse transformation samples in real-world scenarios (*optimization-sensitive samples*), hindering LLMs from learning rich and generalizable optimization strategies.

In this paper, we introduce IR-OptSet, the first public optimization-sensitive dataset for advancing LLM-based IR optimizers. It comprises 170K LLVM IR samples from open-source repositories across 8 representative optimization domains. IR-OptSet defines two core tasks: Code Analysis and Optimized Code Generation, and provides tools for correctness verification, performance evaluation, and dataset expansion. In our experiments, fine-tuning three representative LLMs on IR-OptSet leads to significant accuracy improvements across both tasks. Moreover, the LLM fine-tuned with IR-OptSet *outperforms traditional compiler with the* `-O3` *option* in 64 test cases in terms of performance. Further analysis reveals that IR-OptSet provides greater transformation diversity and representativeness than three widely used IR-oriented datasets, highlighting its potential to drive model-based IR optimization. IR-OptSet is publicly available at https://huggingface.co/datasets/YangziResearch/IR-OptSet.

## 1 Introduction

The rapid advancement of large language models (LLMs) is reshaping software engineering tools [47, 48, 49, 54], driving progress in tasks ranging from code completion [76, 41, 22, 68] to automated program repair [67, 69, 27]. In the domain of compilers, which is one of the most fundamental components in modern computing systems, LLMs have demonstrated promise in enhancing compiler development [74, 73] and testing efficiency [20, 65]. However, one of the compiler's most critical tasks – optimizing code for runtime performance – remains largely reliant on manually crafted, rule-based transformations over intermediate representations (IRs) such as LLVM IR [36]. As compilers grow in complexity, these hand-written rules increasingly struggle with adaptability and scalability, highlighting the need for more flexible, learning-based approaches.

---

*Equal Contribution.

†Corresponding Authors.

39th Conference on Neural Information Processing Systems (NeurIPS 2025) Track on Datasets and Benchmarks.

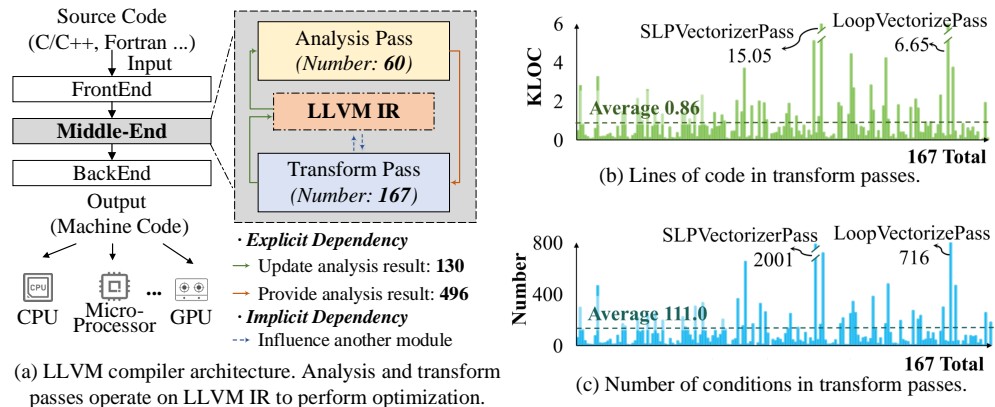

(a) LLVM compiler architecture. Analysis and transform passes operate on LLVM IR to perform optimization.

(b) Lines of code in transform passes.

(c) Number of conditions in transform passes.

Figure 1: Heavy manual efforts in adapting optimization modules in LLVM 19.1.0, the latest released version by the time this work was conducted.

Fig. 1(a) – (c) illustrates the challenges posed by rule-based compiler optimization. Modern compilers such as LLVM [36] apply a long sequence of optimization steps to improve the performance of IR code. At each step, the compiler applies two types of passes (a modular unit for optimization) [38], i.e., the analysis passes and transformation passes to optimize the IR performance. The analysis passes extract optimization-related information from the IR to guide the subsequent transformation pass. For instance, LLVM comprises over 60 analysis passes and 167 transformation passes, many of which exhibit intricate explicit and implicit dependencies (Fig. 1(a)). Each pass encapsulates highly specialized optimization logic; for example, `SLPVectorizerPass`, a key loop optimization transform, spans more than 15,000 lines of code (Fig. 1(b)) and contains over 2,000 conditional branches (Fig. 1(c)). This level of complexity underscores the significant manual effort required to design and maintain these optimization rules.

Recent studies have investigated LLM-based IR optimizers by formulating the task as a generation problem – from unoptimized IR to optimized IR [14, 13]. Correspondingly, several IR-oriented datasets [52, 19, 10, 56, 14] have been proposed to support this emerging direction. While they show early promise in utilizing LLMs to automate IR-level optimizations, they remain fundamentally limited by a lack of samples that reflect how compilers apply optimizations across diverse program structures – that is, these datasets lack *optimization-sensitive* samples whose IR trigger representative and varied transformations in real-world optimization scenarios. As a result, LLMs trained on these datasets struggle to generalize to varying programs and fail to capture the intricate optimization logic that human experts encode into compilers. Our experiments in Sec. 4.2 and Sec. 4.4 empirically validate this limitation and highlight the ample room for improvement.

To bridge this gap, we introduce IR-OptSet, the first public dataset comprising a large number of **optimization-sensitive** samples, designed to enhance LLM-based IR optimizer in comprehending and performing IR-level optimizations. We focus on LLVM IR due to its widespread use [2, 26, 28], as well as its well-defined, human-readable syntax and semantics [36], which make it suitable for language modeling. IR-OptSet comprises 170,564 LLVM IR, sourced from 1,704 open-source GitHub repositories across 8 key compiler optimization target domains. Each domain contains programs tailored to trigger different types of optimization effects, with each program exhibiting an average of 22.89 effective optimization steps. The dataset defines two core tasks aligned with the compiler optimization pipeline: (1) Code Analysis and (2) Optimized Code Generation. To support robust evaluation and future extensibility, IR-OptSet provides three tools: a verification module for validating the correctness of LLM-generated IR without execution, a static performance evaluation tool, and a modular toolchain for scalable dataset expansion.

In the experiment, we selected one representative LLM-based IR optimizer [14] and two general-purpose code LLMs [41, 25] and fine-tuned them using IR-OptSet. Results indicate that IR-OptSet effectively improves the accuracy of all three models across two tasks. Further evaluation using the top-performing model (LLM Compiler) shows that IR-OptSet offers greater transformation diversity and representativeness compared to three widely used IR-oriented datasets [52, 19, 9]. Notably, in 64 cases, the model fine-tuned on IR-OptSet generates optimizations that outperform those produced by LLVM -O3, advancing the potential for model-driven IR optimization approaches.

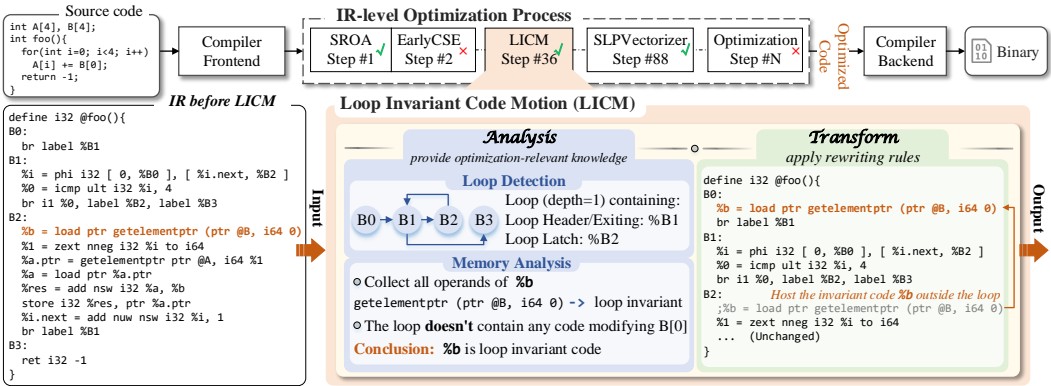

Figure 2: IR-level optimization process in LLVM 19.1.0.

## 2   Background: IR-level Optimization Process

IR-level optimization is a fundamental phase in modern compilers, aimed at improving program performance while preserving semantic correctness. As discussed in Sec. 1, compilers like LLVM implement this process as a sequence of optimization steps, each consisting of two essential passes: the analysis pass and the transform pass [38].

**Analysis Pass.** This pass provides compiler-specific optimization knowledge, capturing how the compiler comprehends the program based on its IR. For instance, in the Loop Invariant Code Motion (LICM) optimization step, as illustrated in Fig. 2, the compiler first performs loop detection to identify loop structures in the IR. It then conducts memory analysis to determine which computations, such as `%b`, remain invariant across loop iterations. The information obtained during analysis directly informs the transform pass by identifying which IR rewrite rules are applicable.

**Transform Pass.** In the transform pass, the compiler performs the rewrite rules guided by the preceding analysis to modify the IR. In the case of LICM of Fig. 2, `%b` is hoisted outside the loop, as determined by the analysis. If no valid rewrite pattern is identified, the IR remains unchanged.

In this paper, we define **Effective Optimization Step** as an optimization step that transforms the IR, such as LICM. The number of effective optimization steps quantifies how much and in what way the IR is transformed through optimization. Such transformations are inherently input-dependent, varying significantly with the characteristics of the input program [63]. This variability poses a major challenge for traditional rule-based compiler design. Even minor IR modifications, such as inserting a benign function call into a loop body in Fig. 2, can hinder handcrafted rules from identifying optimization opportunities, such as recognizing `%b` as loop-invariant. These missed transformations can further affect downstream steps; for example, they may influence SLPVectorizer in detecting vectorizable patterns as a result. As depicted in Fig. 1, adapting to such subtle variations is highly labor-intensive: the `SLPVectorizerPass` alone contains over 15,000 lines of code and more than 2,000 conditional branches to handle diverse cases.

Recent work has explored leveraging LLMs for IR-level optimization inspired by their potential in adapting to IR variations that often challenge rule-based systems [14, 52, 55]. However, these approaches are still constrained by the lack of programs whose IR undergoes diverse transformations during optimization. To address this challenge, we propose IR-OptSet, an optimization-sensitive dataset designed to expose models to diverse and representative transformations. IR-OptSet supports fine-tuning LLMs for IR-level optimization, reducing reliance on handcrafted rules and enabling more adaptive, data-driven optimization strategies.

## 3   IR-OptSet: A Dataset for IR-Level Compiler Optimization

### 3.1   Overview of IR-OptSet

To our best knowledge, IR-OptSet is the first public dataset specifically designed for optimization-sensitive IR-level optimization. Notably, it comprises three key features as outlined below:

Table 1: Composition of source programs and optimization behavior diversity in IR-OptSet.

| Domain | Types of Opt. | #Repos | #LLVM IR | Avg. Eff. Opt. Steps |
|---|---|---|---|---|
| High-Performance Computing (HPC) | Vectorization, parallelism, and memory locality optimizations. | 275 | 17,145 | 23.28 |
| Machine Learning | Vectorization, parallelism, and specialized instruction set utilization (e.g., SIMD). | 95 | 9,366 | 26.64 |
| Multimedia | Vectorization and loop optimizations. | 174 | 15,019 | 22.55 |
| Embedded Systems | Code size reduction, power efficiency, and performance optimization. | 108 | 5,942 | 21.79 |
| System Software | Control flow efficiency, computation, and memory access optimization. | 93 | 6,581 | 20.77 |
| Security | Careful register use and avoidance of side-channel vulnerabilities during optimization. | 94 | 9,252 | 19.92 |
| Reusable Libraries | Inlining and target-specific optimization. | 106 | 7,664 | 20.51 |
| Algorithms | Computation and memory optimizations. | 759 | 99,595 | 27.66 |
| **Total** | - | 1,704 | 170,564 | Avg. 22.89 |

(1) **Optimization-Sensitive.** IR-OptSet captures rich and diverse transformations by focusing on real-world programs that are sensitive to various types of compiler optimizations, ensuring broad coverage and representativeness. It is sourced from 1,704 GitHub repositories spanning 8 domains, as summarized in Table 1. These domains – encompassing representative compiler optimization target programs such as high-performance computing, machine learning, and multimedia – exhibit diverse program-level characteristics that result in rich transformations. In total, IR-OptSet includes 170,564 LLVM IR files, with each program exhibiting an average of 22.89 effective optimization steps. This transformation diversity enables models to learn when and where specific optimizations are applied, advancing research in LLM-based IR optimizer.

(2) **Task-Oriented.** IR-OptSet aligns with the analysis and transform passes of the traditional IR-level optimization process through two tasks: 1) Code Analysis; 2) Optimized Code Generation, aiming to improve LLMs' capabilities in both comprehending and performing IR-level optimizations. To ensure reliability and enable performance assessment, IR-OptSet includes two tools: a verification tool for validating the correctness of model-generated IR, and a static analysis tool for estimating target-specific performance. Together, these tools make IR-OptSet a practical foundation for building and evaluating model-based IR optimization approaches.

(3) **Extensible.** IR-OptSet provides a tool that facilitates dataset expansion and enables exploration of optimization strategies beyond traditional compilers. Users can generate new IR variants by incorporating additional compilers (e.g., AOCC [2], ICX [26]), targeting diverse machines, or applying custom optimization sequences from autotuning and search-based methods [46, 12, 45, 51, 32, 8]. This extensibility enables LLMs to learn more advanced transformations, with the potential to surpass the performance of conventional compiler heuristics.

### 3.2 Data Collection and Pre-processing

The collection and pre-processing of data in IR-OptSet adhere to the following steps:

1. **Code Collection.** We crawled GitHub using the keywords detailed in Appendix A to collect repositories across 8 representative compiler optimization target domains. Repositories that were incomplete or non-compilable were filtered out. The remaining 1,704 repositories were aggregated to create the raw code data.

2. **LLVM IR Extraction.** We extracted LLVM IR immediately after the raw code being processed by the LLVM 19.1.0 frontend, before any optimization passes are applied. By preserving the unoptimized IR, we retain its full potential for analysis and optimization. Following the approach

| Input | Task | | Ground Truth |
|---|---|---|---|
| B0:
**store i32** 0, ptr %0
**br** %B1
B1:
%1 = **load i32**, ptr %0
%2 = **add i32** %1, 1
**store i32** %2, ptr %0
%3 = **icmp slt i32** %2, 10
**br i1** %3, %B1, %B2
B2:
**ret void** | **Code Analysis** | Dominator Tree Construction | [1] %B0
[2] %B1
[3] %B2
Roots: %B0 |
| | | Loop Detection | Loop at depth 1 containing:
%B1\<header>\<latch>\<exiting> |
| | | Memory Access Analysis | ...
*; MemoryUse(3)*
%1 = **load i32**, ptr %0
%2 = **add i32** %1, 1
*; 2 = MemoryDef(3)*
**store i32** %2, ptr %0
... |
| | **Optimized Code Generation** | | B0:
**store i32** 10, ptr %0
**ret void** |

Figure 3: Examples of two tasks in IR-OptSet.

used in LLM Compiler [14, 13], we used `llvm-extract` [34] to split the original IR into new LLVM IR files where each contains a single function. To reduce redundancy, we applied `StructuralHash` [35], a structural hashing technique that computes hashes based solely on the IR's control and data flow structures. This ensures robustness to superficial variations, such as local variable renaming, while preserving semantically relevant features.

3. **IR Normalization.** To further eliminate irrelevant variation and emphasize the semantically meaningful components of LLVM IR, all IR files in IR-OptSet are normalized. This process standardized the naming of basic blocks, local variables, and struct identifiers, and removed comments, empty lines, and extraneous whitespace via an automated script. The normalized IR preserves the original semantics and remains compatible with LLVM 19.1.0, enabling future dataset extension.

4. **Optimization Annotation.** Each IR file in IR-OptSet is annotated with its corresponding effective optimization steps. These annotations capture the IR's sensitivity to different optimization steps. Specifically, we apply the widely adopted `-O3` optimization sequence [39] as a proxy for real-world compiler strategies, and use LLVM's `-print-changed` flag to identify and record the transform passes that modify the IR. To ensure relevance, we filter out IR files where fewer than 8% of the executed passes result in transformations, treating them as optimization-insensitive. This filtering criterion distinguishes IR-OptSet from prior IR datasets and ensures a focus on transformation-rich samples. The final dataset contains 170,564 LLVM IR files.

### 3.3   Tasks in IR-OptSet

Following the IR-level optimization workflow in compilers, we define two tasks: Code Analysis and Optimized Code Generation, as illustrated in Fig. 3. Code Analysis enables models to comprehend IR from an optimization perspective, aiming to extract optimization-relevant knowledge from unoptimized IR to guide transformation decisions. Optimized Code Generation, on the other hand, requires models to directly generate optimized IR from a given unoptimized IR. Data processing steps for each task are detailed in subsequent subsections.

IR-OptSet is designed to enhance language models' ability to interpret and apply IR-level optimizations through fine-tuning, enabling them to perform *semantically correct* transformations – that is, to generate optimized IR that preserves the functional equivalence (i.e., correctness) of the original IR across all valid inputs. We emphasize functional equivalence because it serves as the cornerstone of reliable LLM-based IR optimizer. Building on this principle, IR-OptSet provides foundational infrastructure to support future advances in LLM-driven compilation. It enables a shift from handcrafted rewriting rules to more adaptive, data-driven optimization strategies, paving the way for language models to eventually surpass traditional compilers in optimization performance.

#### 3.3.1   Code Analysis

To capture both control-flow and data-flow aspects [44] of optimization-relevant analysis, we select three representative sub-tasks under Code Analysis: Dominator Tree Construction, Loop Detection,

and Memory Access Analysis. These sub-tasks align with widely adopted analysis passes in LLVM 19.1.0 – `DominatorTreeAnalysis`, `LoopAnalysis`, and `MemorySSAAnalysis` [35]. Each pass generates its analysis results through LLVM's official API in a human-readable textual format, making them well-suited for training and evaluating language models.

**Dominator Tree Construction.** As a fundamental control-flow analysis, dominator tree construction identifies dominance relationships among basic blocks (i.e., straight-line code sequences with a single entry and exit) within a function. Specifically, a block $B_i$ is said to dominate $B_j$ if every execution path from the function's entry to $B_j$ passes through $B_i$[44]. Given the normalized LLVM IR of a function, the model predicts the dominator tree by assigning a hierarchy level to each basic block, as illustrated in Fig. 3. The ground truth is obtained from LLVM's `llvm::DominatorTree::print` API, which implicitly encodes dominance relationships—for example, `%B0` dominates both `%B1` and `%B2`, and `%B1` dominates `%B2`.

**Loop Detection.** As another key control-flow analysis, loop detection identifies loop structures based on the dominator tree information [44]. Given the normalized LLVM IR of a function, the model predicts the role of each basic block within the loop structure, as shown in Fig. 3. The ground truth, obtained via the `llvm::LoopInfo::print` API, specifies that the input function contains a single-level loop composed of `%B1`, which serves as the loop header, latch, and exiting block.

**Memory Access Analysis.** This subtask requires the model to predict memory access relationships using the MemorySSA form provided by `llvm::MemorySSA::print` API, as shown in Fig. 3. Although this subtask belongs to data-flow analysis, it fundamentally relies on control-flow analysis to determine where memory definitions and uses occur and how they propagate across different execution paths [44]. This joint reasoning over control- and data-flow makes the subtask considerably more challenging than tasks based purely on control-flow analysis.

### 3.3.2 Optimized Code Generation

Optimized Code Generation enhances the model's ability to perform IR-level optimizations by tasking it with generating optimized IR from the unoptimized IR, as shown in Fig. 3. Ground truth labels are generated using LLVM's `-O3` optimization option, which reflects a wide range of representative compiler optimization strategies. To ensure consistency and reduce variability across the training data, the resulting optimized IR is further normalized using the procedure outlined in Sec. 3.2.

### 3.4 Toolchain

To support correctness verification, performance evaluation, and dataset extensibility in IR-OptSet, we integrate three complementary tools into IR-OptSet.

**Correctness Verification.** To ensure the correctness of model-generated IR without requiring execution, we adopt a two-stage verification process. First, the LLVM function-level verifier [35] is invoked to ensure that the generated IR conforms to LLVM's structural and syntactic rules. Then, to verify functional equivalence, the IR is passed through Alive2 [40], a widely used formal verification tool that verifies functional equivalence between the original and optimized IR. Since Alive2 applies strict formal verification and may timeout on complex IR, we consider a model-generated IR correct if it exactly matches the ground-truth optimized IR or successfully passes both verification stages.

**Performance Evaluation.** To assess the effectiveness of correctness-verified IR, we perform performance analysis on the target machine. Specifically, we first use the `llc` tool in LLVM to compile the IR into machine-specific assembly code. We then employ `llvm-mca` [34], LLVM's static performance analysis tool, to estimate the execution performance of the generated assembly on the target machines. Optimization effectiveness is quantified by comparing the predicted performance of the optimized IR to the unoptimized version, where greater value indicates more effective optimization.

**Extension.** While IR-OptSet focuses on enhancing models to comprehend and perform correct IR-level optimizations, its extensible toolchain also facilitates exploration of advanced optimization strategies that surpass those of traditional compilers. Users can apply external autotuning methods [34] to discover high-performance sequences and easily integrate new IR variants for continuous dataset expansion and iterative improvement of LLM-based optimizers. Additionally, the toolchain supports generating IR variants across different compilers and target machines, enabling further enhancement of LLM-based optimizers through cross-compiler and cross-machine generalization.

Table 2: Comparison of accuracy across two tasks of three LLMs fine-tuned by IR-OptSet. "Code Anal." refers to Code Analysis, while "Opt. Code Gen." refers to Optimized Code Generation.

| | Code Anal. | | Opt. Code Gen. | | | Code Anal. | | Opt. Code Gen. | | |
|---|---|---|---|---|---|---|---|---|---|---|
| | EM(%) | BLEU | EM(%) | BLEU | Corr.(%) | EM(%) | BLEU | EM(%) | BLEU | Corr.(%) |
| | **Without Fine-Tuning** | | | | | **Fine-Tuned** | | | | |
| **LLM Compiler** | 0.00 | 0.07 | 0.00 | 0.38 | 6.00 | 38.52 | **0.96** | **52.00** | **0.95** | **84.40** |
| **StarCoder2** | 0.00 | 0.03 | 0.00 | 0.08 | 3.80 | **48.10** | 0.85 | 4.80 | 0.70 | 57.40 |
| **Qwen2.5-Coder** | 0.00 | 0.01 | 0.00 | 0.22 | 12.20 | 11.98 | 0.91 | 2.20 | 0.79 | 43.60 |

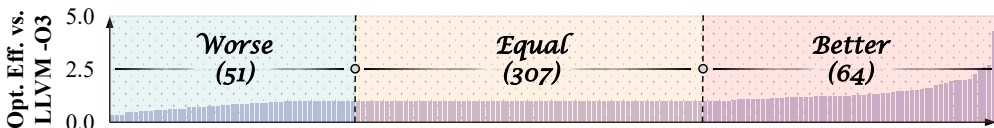

Figure 4: Optimization effectiveness comparison: LLM Compiler FTD IR-OptSet vs. LLVM -O3.

# 4 Experiment

This section addresses the following research questions (RQs):

- **RQ.1**: Can IR-OptSet help LLMs better comprehend and perform IR-level optimizations, even when trained on a small-scale subset of IR-OptSet? (Sec. 4.2)

- **RQ.2**: Can IR-OptSet enable LLMs to generate optimizations that have the potential to surpass traditional compiler-generated results in terms of performance in real-world scenarios? (Sec. 4.3)

- **RQ.3**: Can IR-OptSet provide a more diverse and representative set of transformations compared to existing datasets? (Sec. 4.4)

## 4.1 Experimental Setup

**Fundamental Models.** We selected three LLMs for evaluation: 1) LLM Compiler FTD 7B [14], 2) StarCoder2-3B [41], and 3) Qwen2.5-Coder-1.5B [25]. LLM Compiler FTD 7B is specifically designed for IR-level optimization and serves as a dedicated LLM-based IR optimizer. In contrast, StarCoder2-3B and Qwen2.5-Coder-1.5B are general-purpose code LLMs not explicitly trained for IR optimization. We include them to demonstrate that IR-OptSet supports generalization across different LLM scales and architectures. By fine-tuning these LLMs with IR-OptSet, we show that both dedicated and general-purpose code LLMs can acquire enhanced capabilities to understand and perform IR-level optimizations. All fine-tuned models and code are available at `https://github.com/yilingqinghan/IR-OptSet`.

**Evaluation Metrics.** To evaluate the inference capability of models fine-tuned with IR-OptSet, we adopt standard alignment metrics for Code Analysis, including Exact Match Accuracy (**EM**) and BLEU [50]. For Optimized Code Generation, we use EM and BLEU to assess textual similarity, and additionally introduce two complementary metrics: Correctness (**Corr.**) and Optimization Effectiveness (**Opt. Eff.**). These metrics capture both the accuracy and performance of the generated IR. Higher values across all metrics indicate better results. Detailed definitions of evaluation metrics are provided in Appendix C.

**Training Settings.** All models are fine-tuned and evaluated on a server equipped with a 40-core Intel Xeon Gold 6248 CPU and 2 NVIDIA A100 GPUs (80GB memory each). The fine-tuning objective is sequence-to-sequence prediction, covering both Code Analysis and Optimized Code Generation in IR-OptSet. All models leverage Low-Rank Adaptation (LoRA) [24] for fine-tuning. We configure the LoRA modules with `lora_r=32`, `lora_alpha=16`, and `lora_dropout=0.05`. Target layers are set to {`q_proj, k_proj, v_proj, o_proj`} for LLM Compiler and StarCoder2, and {`qkv_proj, o_proj`} for Qwen2.5-Coder. All models use a batch size of 2 and a learning rate of $1 \times 10^{-4}$.

## 4.2 Accuracy Improvement across Various LLMs

To assess the impact of IR-OptSet, we randomly selected a 6,000-sample subset from IR-OptSet, split into 5,000 for training, 500 for validation, and 500 for testing. Detailed statistics are provided in

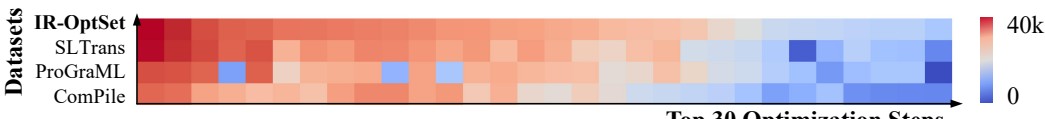

Figure 5: Heatmap analysis of the top 30 most commonly used optimization steps in LLVM 19.1.0 across four datasets. Color intensity represents the frequency of effectiveness.

Table 3: Accuracy of LLM Compiler fine-tuned by IR-OptSet, SLTrans, ProGraML and Compiler for Code Analysis and Optimized Code Generation tasks.

| LLM Compiler | Code Anal. | | | | | | Opt. Code Gen. | | |
| --- | --- | --- | --- | --- | --- | --- | --- | --- | --- |
| | Dom. Tree Const. | | Loop Dete. | | Mem. Anal. | | | | |
| | EM(%) | BLEU | EM(%) | BLEU | EM(%) | BLEU | EM(%) | BLEU | Corr.(%) |
| **FTD IR-OptSet** | **90.60** | **0.99** | **81.60** | 0.94 | **54.00** | **0.92** | **45.40** | **0.86** | **81.00** |
| FTD SLTrans | 78.00 | 0.95 | 73.00 | 0.92 | 22.60 | 0.38 | 40.60 | 0.86 | 75.40 |
| FTD ProGraML | 78.2 | 0.98 | 63.80 | **0.98** | 33.80 | 0.89 | 27.00 | 0.66 | 70.60 |
| FTD ComPile | 73.6 | 0.94 | 62.00 | 0.89 | 38.40 | 0.87 | 43.40 | 0.85 | 76.60 |

Appendix E. Table 2 reports accuracy improvements across the two tasks for three LLMs fine-tuned on this subset. Despite the small training size, all models exhibited notable performance gains. The LLM Compiler, although designed for IR optimization, initially performed poorly on Optimized Code Generation due to limited exposure to diverse and representative transformations. After fine-tuning with IR-OptSet, it showed substantial improvements in both Code Analysis (EM: +38.52%, BLEU: +0.89) and Optimized Code Generation (EM: +52.00%, BLEU: +0.57, Corr: +78.40%). Even not tailored for IR tasks, StarCoder2 and Qwen2.5-Coder also benefited significantly. Notably, StarCoder2 achieved the highest EM score on Code Analysis, demonstrating the effectiveness of IR-OptSet in equipping general-purpose code models with IR-level optimization capabilities.

_Answer to RQ.1_: **IR-OptSet can help LLMs better comprehend and perform IR-level optimizations, even when trained on a small-scale subset of IR-OptSet.**

### 4.3 Improvement over Traditional Compiler

We selected the LLM Compiler fine-tuned with IR-OptSet (LLM Compiler FTD IR-OptSet) from Sec. 4.2 and evaluated its optimization effectiveness on 422 correctness-verified IR samples generated in the Optimized Code Generation task. As a baseline, we compared its output to LLVM 19.0.1 with the -O3 flag (LLVM -O3), representing a widely used real-world optimization configuration. Results are summarized in Fig. 4. LLM Compiler FTD IR-OptSet achieved performance comparable to LLVM -O3 in 307 cases, and notably outperformed it in **64** instances. These results highlight the potential of IR-OptSet to empower LLM-based optimizers to surpass traditional compiler pipelines. A representative case study is shown in Appendix G.

_Answer to RQ.2_: **IR-OptSet can enable LLM to generate optimizations that have the potential to surpass traditional compiler-generated results in terms of performance in real-world scenarios.**

### 4.4 Diversity and Generalization in Transformations

We selected the LLM Compiler – due to its highest average accuracy on both tasks (Sec. 4.2) – to evaluate IR-OptSet against three representative IR-oriented datasets: SLTrans [52], ProGraML [9], and ComPile [19]. To ensure fairness, the LLVM IR from all datasets was normalized using the procedure outlined in Sec. 3.2.

**Diversity.** We randomly sampled 10,000 IR examples from each dataset and conducted a static analysis to assess the diversity of transformations. Specifically, we measured the effectiveness frequency of the top 30 most commonly applied optimization steps in LLVM 19.1.0 across all four datasets, as shown in Fig. 5. Since a single optimization step may be applied multiple times within the -O3 sequence [29], the highest observed frequency reaches 40,552. Fig. 5 reveals that IR-OptSet consistently triggers the highest effectiveness frequency across nearly all top-30 optimization steps, highlighting its broad transformation coverage and superior ability to capture diverse and representative optimization behaviors.

Table 4: Comparison of IR-OptSet with existing IR datasets for compiler optimization. A dash ("-") is used where data is not publicly available.

| Dataset | Samples | Source Repos | Dataset Objective | Toolchain | Avg. Eff. Opt. Steps |
|---------|---------|--------------|-------------------|-----------|----------------------|
| IR-OptSet | 170K | 1,704 | Code Analysis, Optimized Code Generation | Correctness Verification, Performance Evaluation, Extension | 25.50 |
| SLTrans | 6.9M | - | Neural Code Translation | - | 21.92 |
| ProGraML | 469K | - | Code Analysis | - | 13.33 |
| ComPile | 1.9T | - | Code Analysis, Optimized Code Generation | Extension | 10.60 |

**Generalization.** To evaluate the generalization across the four datasets, we fine-tuned the LLM Compiler on 5,000 randomly sampled examples from each dataset. For testing, we randomly selected 125 samples from each dataset, forming a 500-sample test set (with no overlap between the training and test sets). We then assessed the four fine-tuned models' accuracy on Code Analysis and Optimized Code Generation tasks. Detailed data and token statistics are provided in Appendix E.

Table 3 shows that the LLM Compiler fine-tuned with IR-OptSet outperforms other models across nearly all tasks. To further assess generalization, we measured correctness on test samples from all four datasets, as illustrated in Fig. 6. It shows that each fine-tuned model performs best on samples from its own training dataset (e.g., LLM Compiler fine-tuned on ComPile achieves the highest accuracy on ComPile samples). However, the LLM Compiler fine-tuned on IR-OptSet consistently outperforms all other fine-tuned models on datasets outside their own (e.g., LLM Compiler fine-tuned on IR-OptSet achieves the second-highest accuracy on ComPile samples), demonstrating that IR-OptSet more effectively exposes the model to representative and diverse transformations, leading to broader generalization.

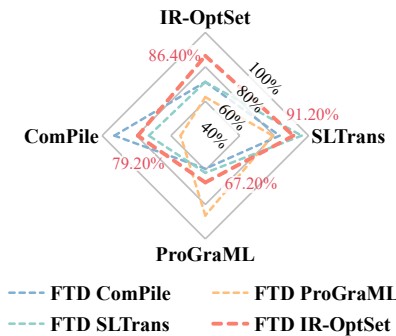

Figure 6: Generalization of 4 datasets.

***Answer to RQ.3***: **IR-OptSet can provide a more diverse and representative set of transformations compared to existing datasets.**

## 5 Related Work

**Datasets for Compiler Optimization.** The rise of machine learning in compiler research has underscored the importance of high-quality datasets. IR-oriented datasets [52, 19, 10, 56, 14], earlier high-level language datasets [53, 41, 42, 3, 16], and automatic dataset generation tools [33, 66, 6, 11] have advanced data-driven approaches to compiler optimization. Table 4 compares IR-OptSet with representative IR datasets across key dimensions, covering the number of samples, source repositories, dataset objectives, provided toolchains, and the average effective optimization steps. Although IR-OptSet contains fewer samples, it achieves a substantially higher average effective optimization steps, indicating richer optimization behavior per sample and underscoring IR-OptSet's value as an optimization-sensitive dataset that fundamentally differs from existing datasets.

**Large Language Models for Compilers.** LLMs have made significant progress in code understanding and generation, especially in code-related research [31, 21, 62, 30, 61, 1, 64, 70]. Several multilingual models have included LLVM IR in their training data [41, 17, 76, 52]. Approaches like LLM Compiler [13, 14] have demonstrated the potential of IR-to-IR optimization. Prior work [74, 73, 75, 72], such as BePilot and VEGA, further enhances compiler backend development efficiency to streamline the overall compiler design process.

**AI for Compilation Optimization.** Automatic tuning has been a key research direction in compiler community [46, 12, 45, 51, 32, 8], aiming to automate the selection and sequencing of optimization passes. Building on this foundation, recent deep learning frameworks [60, 15, 5, 7, 58, 4] have been proposed to enhance compiler workflows. Machine learning has also advanced parallel workload

optimization, improving task scheduling, vectorization, and code generation [57, 71, 43, 23]. Additionally, the growing interest in learning representations of compiler IR [18, 59, 10] has driven improvements in optimization prediction, code similarity, and performance modeling.

## 6 Discussion

**Limitation.** One limitation of IR-OptSet lies in the fact that IR code can exceed the token context window of current LLMs. To address this, we split each LLVM IR file into shorter snippets, each corresponding to a single IR function – a widely adopted strategy in existing IR datasets [14]. While this approach restricts cross-function optimizations, the impact can be mitigated through Link Time Optimization (LTO)[37], which enables whole-program analysis at the linking stage.

**Broader Impact.** While IR-OptSet is primarily designed to help LLMs better comprehend and perform IR-level optimizations, its influence potentially extends far beyond IR-level optimizations. Prior work such as IRCoder [52] and TransCoder-IR [55] has demonstrated that incorporating IR into training datasets can strengthen models' robustness to prompts and improve their multilingual code completion, code understanding, and instruction-following capabilities. Building upon these insights, introducing IR-level optimization tasks in model training may not only advance optimization performance but also foster more generalizable and resilient code understanding across a wide range of programming scenarios.

**Potential Societal Impact.** IR-OptSet does not contain any personally identifiable information or offensive content, thereby mitigating any potential negative societal impact.

**Conclusion.** In this paper, we present IR-OptSet, the first public optimization-sensitive dataset for advancing LLM-based IR optimizers. It contains 170,564 LLVM IR samples from 1,704 open-source repositories across 8 optimization domains, with each program exhibiting an average of 22.89 effective optimization steps. IR-OptSet defines two core tasks – Code Analysis and Optimized Code Generation– to enhance LLMs' understanding and generation of IR-level optimizations. Experiments show that IR-OptSet consistently boosts model accuracy and exposes more diverse, representative transformations than existing datasets, enabling LLM-generated IR to outperform traditional compiler outputs and supporting future progress in model-driven optimization.

## Acknowledgement

We would like to thank all anonymous reviewers for their insightful feedback. This work was supported by National Key R&D Program of China, Grant No.2024YFB4505701. This work was also supported by the National Natural Science Foundation of China, Grant No.U23B2020, No. 62090024, No. 62302479, No.62232015.

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

## A  Keywords for GitHub Repository Crawling

Table 5 presents the keywords used for GitHub repository crawling. After applying appropriate filtering, 1,704 repositories are reserved as the source of high-level language code for IR-OptSet.

Table 5: Keywords used for GitHub repository crawling.

| Domain | Keywords |
| --- | --- |
| High-Performance Computing (HPC) | Finite Element Solver, Geometry Simulation, Geometry Solver, Graphics Simulation, Graphics Solver, Molecular Dynamics, Multiphysics Simulation, Network Simulation, Optimization Simulation, Optimization Solver, Parallel Computing, Physics Simulation, Physics Solver, Plasma Physics, Quantum Chemistry, Robotics Solver, Security Solver, Simulation, Computational Fluid, Computational Fluid Dynamics, Numerical Linear Algebra |
| Machine Learning | Machine Learning |
| Multimedia | Computer Graphics, Graphics Algorithm, Graphics Engine, Image Processing, Audio Engine |
| Embedded Systems | Control Systems, Device Driver, Embedded, Robotics, Robotics Engine |
| System Software | Filesystem, Garbage Collection, JIT Runtime, Operating System, Virtualization, Compiler |
| Security | Cryptography, Security Algorithm, Security Engine |
| Reusable Libraries | Database Engine, Geometry Engine, Optimization Engine, Physics Engine |
| Algorithms | Concurrency, Data Structures, Geometry Algorithm, LeetCode, Network Algorithm, Networking Stack, Numerical Linear, Numerical Methods, Optimization, Optimization Algorithm, Physics Algorithm, Real-time System, Signal Processing, Sorting, ACM ICPC, Algorithm, AtCoder, Benchmark, Codeforces, Competitive, Compression, Competitive Programming |

## B  Task Description

This appendix gives an end-to-end example, allowing readers to see how each ground-truth in Fig. 3 is derived from the unoptimized IR.

### B.1  Example Input

We use the same normalized LLVM IR shown in the left column of Fig. 3. The function contains three basic blocks labeled B0, B1, and B2, where each block is a straight-line sequence of instructions ending with a branch or return. The entry block is B0, and the complete input IR is as follows:

```
B0:
  store i32 0, ptr %0
  br %B1
B1:
  %1 = load i32, ptr %0
  %2 = add i32 %1, 1
  store i32 %2, ptr %0
  %3 = icmp slt i32 %2, 10
  br i1 %3, %B1, %B2
B2:
  ret void
```

This forms a simple loop: B0 initializes %0 to zero, B1 repeatedly increments the value stored in %0, and B2 returns once the value reaches 10. The control-flow graph (CFG) contains three edges: B0→B1, B1→B1 (the loop back-edge), and B1→B2.

### B.2  Dominator Tree Analysis

Starting from the entry block B0, any execution path must pass through B0; therefore, it dominates all other blocks. The only way to reach B1 is through the edge B0→B1, making B0 the immediate

dominator of B1. Similarly, B2 is reachable only via B1→B2, so B1 is its immediate dominator. The resulting dominator tree is a simple chain: B0→B1→B2. Consequently, LLVM's dominator tree analysis reports "`[1] %B0, [2] %B1, [3] %B2` with root `%B0`".

### B.3  Loop Detection Analysis

The CFG contains a back-edge B1→B1, which defines a natural loop. The loop header is B1, as it dominates the loop body; B1 is also the latch block because it contains the back-edge. Additionally, B1 serves as the exiting block because it can branch to B2. Consequently, LLVM's loop analysis reports "Loop at depth 1 containing %B1 <header><latch><exiting>".

### B.4  Memory Access Analysis

The first instruction, "`store i32 0, ptr %0`" in B0, creates the initial memory definition for %0. At the entry of B1, MemorySSA inserts a `MemoryPhi` for %0 to merge the incoming definitions from B0 and the back-edge B1→B1. The `load` in B1 is a `MemoryUse` of that phi node. The next instruction, "`%2 = add i32 %1, 1`", performs an arithmetic operation and does not access memory. The subsequent instruction, "`store i32 %2, ptr %0`", creates a new `MemoryDef`; along the back-edge, this definition becomes one of the inputs to the phi node in the next iteration. In Fig. 3, these appear as `MemoryPhi(%0)` at the top of B1, `MemoryUse(phi)` next to the `load`, and `MemoryDef(%0)` next to the `store`.

### B.5  Optimized Code Generation

Given that B0 initializes %0 to zero and B1 increments its value until it reaches 10, LLVM's optimization directly stores 10 to %0, eliminating the loop. Therefore, the optimized IR becomes:

```
B0:
  store i32 10, ptr %0
  ret void
```

## C  Metrics

**Correctness (Corr.).** Correctness refers to the number of model-generated IRs that pass the verification tool in IR-OptSet, relative to the total number of generated IRs. Specifically, it involves a two-stage check outlined in Sec. 3.4. Formally, Correctness is calculated as:

$$Corr = N_{corr}/N_{total}$$

**Optimization Effectiveness (Opt. Eff.).** Optimization Effectiveness refers to the average performance of the optimized IR against the unoptimized version. Specifically, we use `llc` on an AMD Ryzen 9 7950X 16-Core Processor to generate assembly, then evaluate its execution cycles with `llcm-mca`. For each $c_i$, $p_i$ is calculated as:

$$p_i = (1/t_i)$$

where $t_{ij}$ is the predicted execution cycles of $c_i$.

Optimization Effectiveness is calculated as:

$$OptEff = \sum_{i=1}^{N_{total}} (p_i^{opt}/p_i^{unopt})/N_{total}$$

## D  Hyperparameters and Input/Output Sequence Length Settings

In Table 6, we provide all hyperparameter settings. In this work, we set the context window to 4K, meaning the total input and output must not exceed 4K due to resource limitations.

Table 6: Hyperparameter settings.

| Model | LoRA rank ($r$) | LoRA $\alpha$ | LoRA dropout | Batch size | Learning rate | Target modules |
|-------|-----------------|---------------|--------------|------------|---------------|----------------|
| LLM Compiler | 32 | 16 | 0.05 | 2 | $1 \times 10^{-4}$ | {q_proj, k_proj, v_proj, o_proj} |
| StarCoder2 | 32 | 16 | 0.05 | 2 | $1 \times 10^{-4}$ | {q_proj, k_proj, v_proj, o_proj} |
| Qwen2.5-Coder | 32 | 16 | 0.05 | 2 | $1 \times 10^{-4}$ | {qkv_proj, o_proj} |

# E    Token Statistic

To accommodate the 4K context window, we perform random sampling across all tasks in the datasets, retaining only those samples that fit within the context window. Table 7 shows the data statistics about the number and token of Optimized Code Generation and 3 subtasks of Code Analysis for IR-OptSet, SLTrans, ProGraML, and ComPile.

Table 7: Data statistics about the number and token of Optimized Code Generation and 3 subtasks of Code Analysis for IR-OptSet, SLTrans, ProGraML, and ComPile.

| Task | Metric | IR-OptSet | | | SLTrans | | | ProGraML | | | ComPile | | |
|------|--------|-----------|------|------|---------|------|------|----------|------|------|---------|------|------|
| | | Train | Val. | Test | Train | Val. | Test | Train | Val. | Test | Train | Val. | Test |
| Optimized Code Generation | Total Token | 11.36M | 1.10M | 1.13M | 9.57M | 0.95M | 0.95M | 9.00M | 0.90M | 0.90M | 8.87M | 0.91M | 0.88M |
| | Average Token | 2,253.9 | 2,203.2 | 2,261.1 | 1,914.1 | 1,914.0 | 1,904.3 | 1,800.3 | 1,803.8 | 1,801.0 | 1,773.9 | 1,825.5 | 1,755.0 |
| | Median Token | 2,292 | 2,243.5 | 2,294 | 1,964 | 1,942 | 1,910 | 1,730.0 | 1,733.5 | 1,770.5 | 1,692 | 1,800.5 | 1,676.5 |
| | Min Token | 997 | 1,157 | 1,162 | 776 | 806 | 868 | 678 | 810 | 785 | 649 | 838 | 776 |
| | Max Token | 3,401 | 3,146 | 3,288 | 3,153 | 3,173 | 3,095 | 3,459 | 3,230 | 3,580 | 3,426 | 3,423 | 3,362 |
| Dominator Tree Construction | Total Token | 6.47M | 0.66M | 0.65M | 5.33M | 0.54M | 0.53M | 5.46M | 0.55M | 0.53M | 3.69M | 0.38M | 0.35M |
| | Average Token | 1,293.1 | 1,320.7 | 1,291.7 | 1,066.4 | 1,076.3 | 1,071.4 | 1,091.4 | 1,119.2 | 1,068.9 | 737.5 | 764.7 | 706.3 |
| | Median Token | 1,257 | 1,306.5 | 1,268 | 986.5 | 1,007.5 | 1,004 | 959 | 1,023.5 | 894 | 575 | 628 | 532.5 |
| | Min Token | 175 | 356 | 224 | 95 | 112 | 96 | 79 | 93 | 84 | 36 | 36 | 98 |
| | Max Token | 2,714 | 2,576 | 2,560 | 2,986 | 2,916 | 2,728 | 3,371 | 3,074 | 2,996 | 2,980 | 2,352 | 2,685 |
| Loop Detection | Total Token | 4.25M | 0.43M | 0.42M | 5.35M | 0.52M | 0.52M | 5.51M | 0.60M | 0.53M | 3.92M | 0.39M | 0.36M |
| | Average Token | 851.5 | 864.8 | 849.8 | 1,069.9 | 1,056.2 | 1,044.1 | 1,101.3 | 1,193.3 | 1,063.1 | 783.5 | 774.5 | 721.2 |
| | Median Token | 854 | 876 | 838 | 971.5 | 897.5 | 1,010.5 | 998 | 1,061.5 | 953.5 | 717 | 687 | 596.5 |
| | Min Token | 174 | 272 | 263 | 71 | 115 | 110 | 67 | 100 | 96 | 36 | 87 | 36 |
| | Max Token | 1,575 | 1,511 | 1,563 | 3,046 | 2,766 | 2,762 | 3,327 | 3,151 | 3,124 | 2,997 | 2,920 | 2,501 |
| Memory Access Analysis | Total Token | 8.52M | 0.84M | 0.84M | 6.50M | 0.63M | 0.64M | 6.19M | 0.64M | 0.63M | 5.01M | 0.53M | 0.49M |
| | Average Token | 1,704.8 | 1,686.4 | 1,678.7 | 1,300.2 | 1,271.6 | 1,285.9 | 1,237.3 | 1,293.1 | 1,265.8 | 1,002.7 | 1,059.7 | 983.2 |
| | Median Token | 1,704.5 | 1,681.5 | 1,633.0 | 1,280 | 1,222 | 1,273.5 | 1,143 | 1,245 | 1,224 | 819 | 870 | 769 |
| | Min Token | 259 | 348 | 194 | 74 | 150 | 99 | 69 | 110 | 82 | 37 | 37 | 89 |
| | Max Token | 3,178 | 2,986 | 3,007 | 3,030 | 2,914 | 3,010 | 3,275 | 3,185 | 3,152 | 3,002 | 2,804 | 2,998 |

# F    Analysis of Error and Performance Improvement Patterns

## F.1    Error Patterns

We conducted an in-depth analysis of the 78 failed cases made by the LLM Compiler in the Optimized Code Generation task as shown in Table 2, and identified three common error patterns. These errors mainly stem from the model's inaccurate reasoning about data flow – that is, how values are defined, propagated, and used across different branches of the program.

- **Define-use error (21 cases).** The model fails to ensure all variables are defined before use. In the example below, %7 is used but never defined.

  ```
  %6 = add i32 %4, %5
  %8 = mul i32 %6, %7
  ```

- **Structural violations in PHI nodes (18 cases).** The model incorrectly constructs PHI node structures. In the example below, %B3 is not a predecessor of %B2.

```
; B2 has two predecessors: B1 and B4
B2:
%5 = phi i32 [ 0, %B1 ], [ %10, %B3 ]
```

- **Data type error (7 cases).** The model produces type mismatches between variable definitions and uses. In the example below, %10 is defined as i32, but used as double.

```
%10 = add i32 %0, 1
%11 = fcmp ult double %8, %10
```

## F.2 Performance Improvement Pattern

We further analyzed the 64 cases in Fig. 4 where the LLM outperforms the traditional compiler and categorized them into three common patterns. Note that some cases fall into multiple categories, collectively contributing to their performance gains over LLVM; as a result, the total number of cases in the table below exceeds 64. These patterns highlight scenarios in which the LLM exhibits stronger optimization capabilities.

- **Rewriting complex conditions into simple ones (32 cases).** The example below shows how the LLM simplifies a switch statement into a simple conditional branch.
  LLVM:

```
switch i32 %0, label %B1 [
 i32 0, label %B1
 i32 1, label %B2
 ]
```

  LLM Compiler FTD IR-OptSet:

```
%1 = icmp eq i32 %0, 1
br i1 %1, label %B2, label %B1
```

- **Removing unnecessary computations (24 cases).** The example below demonstrates constant folding across multiple operations.
  LLVM:

```
%84 = add nsw i64 %i, 1
%85 = add nsw i64 %84, 2
```

  LLM Compiler FTD IR-OptSet:

```
%85 = add nsw i64 %i, 3
```

- **Replacing verbose code with built-in functions.** For example, a loop that zeros out an array can be replaced with a built-in memset function.
  LLVM (a loop that writes zero values to an array):

```
loop:
%i = phi i64 [ 0, %entry ], [ %next, %loop ]
%ptr = getelementptr <2 x float>, ptr %p, i64 %i
store <2 x float> 0.0, ptr %p
%next = add i64 %i, 2
%cond = icmp ult i64 %next, %size
br i1 %cond, label %loop, label %exit
```

  LLM Compiler FTD IR-OptSet:

```
call void @llvm.memset.p0.i64(ptr %p, i8 0, i64 %size)
```

# G Case Study

We provide a representative case study that compares the LLM Compiler FTD IR-OptSet with the IR produced by traditional compilation (LLVM -O3). In this IR sample, the LLM-generated IR is structurally simpler and results in better execution performance relative to the LLVM -O3 baseline. This case illustrates the practical potential of model-driven compilation to complement or even outperform traditional optimization pipelines.

## G.1 IR Comparison

The following shows the code optimized by LLVM -O3, and the code optimized by LLM Compiler FTD IR-OptSet. Both optimization results can be verified for correctness. The differences between these two optimization results are highlighted in blue.

**Optimized IR (LLVM -O3)**

```
define dso_local void @case(i32 noundef %0, i32 noundef %1){
B0:
tail call void @llvm.memset.p0.i64(ptr @rs, i8 0, i64 16, i1 false)
%2 = icmp eq i32 %0, 0
br i1 %2, label %B1, label %B4
B1:
switch i32 %1, label %B4 [
i32 0, label %B3
i32 1, label %B2
]
B2:
br label %B3
B3:
%3 = phi i8 [ 0, %B2 ], [ 1, %B1 ]
store i8 %3, ptr getelementptr inbounds nuw (i8, ptr @rs, i64 16), align 16
br label %B4
B4:
ret void
}
```

**Optimized IR (LLM Compiler FTD IR-OptSet)**

```
define dso_local void @case(i32 noundef %0, i32 noundef %1, i32 noundef %2, i32 noundef %3){
B0:
tail call void @llvm.memset.p0.i64(ptr @rs, i8 0, i64 16, i1 false)
%7 = icmp eq i32 %0, 0
br i1 %7, label %B1, label %B3
B1:
%8 = icmp eq i32 %1, 0
br i1 %8, label %B2, label %B3
B2:
store i8 1, ptr getelementptr inbounds (i8, ptr @rs, i64 16), align 4
br label %B3
B3:
ret void
}
```

### G.2 Assembly Comparison.

The corresponding x86 assembly code generated by `llc` on AMD Ryzen 9 7950X 16-Core Processor for both the LLM-predicted IR and the `-O3` IR is shown as below.

```
Assembly Code
(LLVM -O3)

# %bb.0:
    xorps xmm0, xmm0
    movaps xmmword ptr [rip + rs], xmm0
    test edi, edi
    je .LBB0_1
.LBB0_6:
    ret
.LBB0_1:
    test esi, esi
    je .LBB0_2
# %bb.3:
    cmp esi, 1
    jne .LBB0_6
# %bb.4:
    xor eax, eax
    mov byte ptr [rip + rs+16], al
    ret
.LBB0_2:
    mov al, 1
    mov byte ptr [rip + rs+16], al
    ret
```

```
Assembly Code
(LLM Compiler FTD IR-OptSet)

# %bb.0:
    xorps xmm0, xmm0
    movaps xmmword ptr [rip + rs], xmm0
    test edi, edi
    je .LBB0_1
.LBB0_3:
    ret
.LBB0_1:
    test esi, esi
    jne .LBB0_3
# %bb.2:
    mov byte ptr [rip + rs+16], 1
    ret
```

### G.3 Performance Evaluation through the Performance Evaluation Tool

To quantify the performance implications of the observed structural differences, we utilize the performance evaluation tool in IR-OptSet to simulate the generated x86 assembly. This tool estimates performance metrics, including instruction throughput and total execution cycles, based on processor micro-architectural models. The performance of the two versions of assembly code is calculated with a default repetition count of 100 iterations.

### G.4 Analysis

- **IR-level Comparison.** The LLVM -O3 optimized IR uses `switch` and `phi` instruction as shown below to handle the second conditional:

```
B1:
  switch i32 %1, label %B4 [
    i32 0, label %B3
    i32 1, label %B2
  ]
B3:
  %8 = phi i8 [1, %B1], [0, %B2]
  br label %B4
```

The LLM Compiler FTD IR-OptSet instead use the `icmp` and `br` instruction:

```
B1:
  %8 = icmp eq i32 %1, 0
  br i1 %8, label %B2, label %B3
```

This removes one basic block and the `phi` instruction, yielding a flatter control-flow graph.

- **Downstream Assembly Comparison.** Because the `phi` instruction is gone, the compiler no longer needs to synthesize zero via `xor` or merge two values, and the `switch`-based jump table disappears. Consequently, the x86 output drops from 15 instructions to 9 instructions, directly reducing branch-prediction pressure and data-move overhead.

- **Performance Improvement.** The performance evaluation tool (100 iterations) predicts 257 cycles for the model's code versus 309 cycles for the `-O3` baseline – a 17 % reduction that correlates precisely with the removed IR and assembly instructions.

## H   Effectiveness Frequency

Table 8 lists the effectiveness frequency of the top 30 most commonly used optimization steps.

Table 8: Effectiveness frequency of the top 30 most commonly used optimization steps across IR-OptSet, SLTrans, ProGraML, and ComPile.

| Optimization Step Name | IR-OptSet | SLTrans | ProGraML | ComPile |
|---|---|---|---|---|
| SimplifyCFG | 40552 | 38601 | 21079 | 15381 |
| InstCombine | 32406 | 30077 | 19948 | 13892 |
| LoopSimplify | 21243 | 22975 | 16049 | 6109 |
| LCSSA | 16106 | 20055 | 16691 | 3773 |
| LoopRotate | 16887 | 16270 | 167 | 4919 |
| GlobalOpt | 10889 | 10248 | 5345 | 9663 |
| EarlyCSE | 11496 | 7208 | 4887 | 6542 |
| SROA | 10311 | 9558 | 256 | 9491 |
| InferFunctionAttrs | 9465 | 7742 | 6127 | 6074 |
| IndVarSimplify | 12420 | 8388 | 4521 | 3246 |
| LICM | 12864 | 4615 | 2044 | 4332 |
| JumpThreading | 7527 | 7420 | 4629 | 2680 |
| TailCallElim | 7747 | 5867 | 411 | 7401 |
| PostOrderFunctionAttrs | 6453 | 3871 | 5118 | 4724 |
| LoopUnroll | 6178 | 6851 | 4197 | 1777 |
| GVN | 5409 | 5057 | 3747 | 1478 |
| IPSCCP | 5028 | 2511 | 3450 | 2396 |
| InstSimplify | 3411 | 4211 | 3342 | 759 |
| Reassociate | 3651 | 3462 | 1780 | 865 |
| CorrelatedValuePropagation | 3853 | 1948 | 1500 | 1563 |
| LoopSimplifyCFG | 2958 | 960 | 1850 | 667 |
| LoopInstSimplify | 1773 | 902 | 1005 | 563 |
| InferAlignment | 1260 | 759 | 876 | 365 |
| SCCP | 794 | 471 | 509 | 170 |
| BDCE | 606 | 543 | 324 | 126 |
| DSE | 690 | 251 | 149 | 355 |
| LoopLoadElimination | 553 | 345 | 408 | 112 |
| LoopVectorize | 551 | 329 | 410 | 111 |
| SLPVectorizer | 726 | 55 | 351 | 232 |

## I   Prompt Template

We adopt a unified prompt format to support various LLVM IR-related tasks. Each prompt begins with a task instruction, followed by the corresponding IR input enclosed in tags. Figure 7 shows examples for both optimization and analysis tasks.

```
[INST]Optimize the following LLVM IR with O3:
(Unopt LLVM IR)
[\INST]
Opt IR:
(Opt LLVM IR)
```

```
[INST]Analyze Dominator Tree / Loops / MemorySSA Walker of the following LLVM IR:
(Unopt LLVM IR)
[\INST]
(Correspondence Analysis)
```

Figure 7: Instruction-based prompt formats used for model fine-tuning. The first example focuses on IR optimization, while the second illustrates structural analysis tasks.

# J Main ToolChain

## J.1 Environment Setup

Our toolchain is built from source to ensure compatibility with LLVM IR formats and Alive2's TV solver. First, build a clean LLVM 19.1.0 release:

```
Build LLVM 19.1.0

cmake -G Ninja ../llvm \
  -DCMAKE_BUILD_TYPE=Release \
  -DLLVM_ENABLE_PROJECTS="clang;lld" \
  -DLLVM_ENABLE_RTTI=ON \
  -DLLVM_ENABLE_EH=ON \
  -DLLVM_TARGETS_TO_BUILD="host"
```

Next, build Alive2 with the TV mode enabled so we can formally verify IR equivalence:

```
Build Alive2 19.0

cmake -G Ninja
  -DCMAKE_PREFIX_PATH=<llvm-installed-path> \
  -DBUILD_TV=1 \
  -DCMAKE_BUILD_TYPE=Release \
  ../llvm-project/alive2
```

## J.2 Tool Usage

Here we describe the functionality of several key toolchain scripts, which utilize the LLVM and Alive2 environments configured above. All example invocation commands are listed below. For the most up-to-date usage instructions, please visit our GitHub page: `https://github.com/yilingqinghan/IR-OptSet`.

### J.2.1 Correctness Verification

Firstly use the `opt_verify.py` script performs batch function-level verification of LLVM IR files (`.ll`/`.bc`) in a directory. Its primary goal is to detect syntax and semantic errors before any downstream compilation or analysis.

**Purpose:** Verifying the well-formedness of IR is a necessary precondition for compilation and ensures that only syntactically valid files proceed to later stages in the toolchain.

```
opt_verify.py

python opt_verify.py \
  --folder ./ir_files \
  --opt-path /path/to/opt \
  --log-errors \
  --log-dir ./opt_error_logs \
  --num-workers 4 \
  --clean
```

Then use the `alive2.py` script accepts paired LLVM IR files – typically the model prediction and the corresponding compiler-generated output – and determines whether the two are semantically equivalent using Alive2's translation validation (TV) engine. Specifically, it automatically extracts two function bodies from each file, combines them into a single valid Alive2-compatible IR module, and runs the equivalence check. It then reports the final success/failure statistics.

**Note:** Alive2 may encounter issues such as SMT solver timeouts, incompatibility with newer LLVM versions (e.g., errors like `ERROR: Unsupported instruction`), lack of interprocedural optimization (IPO) support, and incomplete modeling of certain instructions or memory behaviors.

**alive2.py**

```
python alive2.py \
  --input-dir ./ir/before \
  --output-dir ./ir/verified \
  --suffix ".model.predict.ll" \
  --pure
```

### J.2.2 Performance Evaluation

The `mca_cycles.py` script processes all LLVM IR files in a given directory by lowering them to assembly code using `llc`, then feeds the output into `llvm-mca` to estimate static performance metrics such as total cycle count or block throughput. The backend CPU microarchitecture (e.g., znver3) and dispatch width are configurable. Results are saved as a structured CSV report for analysis.

**mca_cycles.py**

```
python mca_cycles.py \
  --csv results.csv \
  --suffix .ll \
  --from-predict \
  --workers 8 \
  --llc /path/to/llc \
  --llvm-mca /path/to/llvm-mca \
  --mcpu znver3 \
  --dispatch-width 6
```

### J.2.3 Dataset Expansion

The textttcli-frontend.py script serves as a convenient wrapper to apply a sequence of LLVM passes to either source code (e.g., C/C++) or existing IR files. It handles the full pipeline of invoking `clang` for frontend compilation, running `opt` for optimization, managing I/O, and recording pass logs.

**Features:** The script supports automatic sampling, configurable preprocessing strategies, and optional IR cleanup routines. It is designed to streamline batch IR generation and logging, while remaining extensible for custom workflows.

**cli-frontend**

```
python cli-frontend.py pipeline \
  --ir-dir "dataset" \
  --compile-out "UNOPT" \
  --extract-dir "EX" \
  --opt-out "OPT" \
  --opt-flags "-passes='print<loops>' -S" \
  --sample-size 100 \
  --seed 100 \
  --rules strip_all loops_analysis \
  --where all \
  --log-out "LOG" \
  --pre-out "PRE_EX" \
  --post-out "PRE_OPT"
```

The `analyze_changed.py` script analyzes the textual logs produced by LLVM `opt -print-changed` to determine which passes actually caused changes to the IR. By parsing markers in the logs, it distinguishes between effective optimization steps and those which donnot modify the IR.

**Functionality:** It can process all available logs or a user-defined random sample, and summarizes the activity of each pass into a CSV file. This is useful for understanding which transformations are active in practice and for generating statistics or visualizations of optimization effectiveness.

```
python analyze_changed.py \
  -i ./logs \
  --csv pass_summary \
  --sample-size 100 \
  --seed 42
```

This `cli-backend.py` script finally processes the optimized IR files and corresponding logs to construct the final datasets used in model training and evaluation.

**Capabilities:** It supports configurable filters to exclude unsuitable examples (e.g., duplicated samples, sequences exceeding token limits), and formats the data according to predefined templates. This ensures that the resulting dataset is clean, consistent, and ready for downstream fine-tuning or benchmarking.

cli-backend.py

```
python cli-backend.py \
  --pre-dir "PRE_EX" \
  --post-dir "PRE_OPT" \
  --log-dir "LOG" \
  --filters "token_limit_v1" \
  --vfilters dedupe_content \
  --out-dir "FINAL" \
  --make-dataset \
  --train-size 5000 \
  --test-size 500 \
  --valid-size 500 \
  --seed 50 \
  --prompt-template "[INST]Analyze Dominator Tree of the following LLVM IR:
                    \n{pre_ir}[/INST]\n\n{log}\n" \
  --dataset-output "domtree" \
  --token-limit 4000
```

## K  Cross-Dataset Generalization

Table 9: Correctness of models respectively fine-tuned (FTD) on IR-OptSet, ComPile, ProGraML and SLTrans on the test samples from all four datasets.

| FTD Dataset | IR–OptSet | ComPile | ProGraML | SLTrans |
|---|---|---|---|---|
| FTD IR–OptSet | **0.864** | 0.792 | 0.672 | 0.912 |
| FTD ComPile | 0.712 | **0.928** | 0.592 | 0.832 |
| FTD ProGraML | 0.624 | 0.544 | **0.864** | 0.792 |
| FTD SLTrans | 0.712 | 0.728 | 0.616 | **0.960** |

## L  Confidence Intervals

We conducted three independent runs to assess the result consistency of the Optimized Code Generation task, and calculate the confidence interval. The results, summarized in Table 10, demonstrate stable performance across runs. Reported values represent the mean ± 95% confidence interval.

Table 10: Consistency analysis for the Optimized Code Generation task.

| Model | EM(%) | BLEU | Corr(%) |
|---|---|---|---|
| LLM Compiler FTD IR-OptSet | 53.80 ± 4.07 | 0.96 ± 0.04 | 85.33 ± 2.24 |
| StarCoder2 FTD IR-OptSet | 4.60 ± 1.31 | 0.68 ± 0.05 | 56.20 ± 4.74 |
| Qwen2.5-Coder FTD IR-OptSet | 3.00 ± 1.79 | 0.79 ± 0.02 | 44.00 ± 2.17 |

## M Reproducibility and Compute Requirements

All experiments can be reproduced using the released scripts and settings. On a machine equipped with two NVIDIA A100 GPUs, fine-tuning for the **Code Analysis** typically takes 2–3 hours, and its evaluation takes 0.5–2 hours. For the **Optimized Code Generation** task, fine-tuning generally requires 3–6 hours, while evaluation takes approximately 2–4 hours.

## N Licenses for existing assets

Table 11: Licenses and Sources for Assets Used in This Work

| Asset | Version | License | URL / Source |
|---|---|---|---|
| LLVM | 19.1.0 | Apache 2.0 | `https://github.com/llvm/llvm-project` |
| Alive2 | 19.0 | MIT | `https://github.com/AliveToolkit/alive2` |
| LLM Compiler FTD | 7B | Meta LLM Compiler License | `https://huggingface.co/facebook/llm-compiler-7b-ftd` |
| StarCoder2 | 3B | BigCode OpenRAIL-M v1 | `https://huggingface.co/bigcode/starcoder2-3b` |
| Qwen2.5-Coder | 1.5 | Apache 2.0 | `https://huggingface.co/Qwen/Qwen2.5-Coder-1.5B` |
| IR-OptSet | v1.0 | CC BY 4.0 | `https://huggingface.co/datasets/YangziResearch/IR-OptSet` |
| SLTrans | — | CC BY SA 4.0 | `https://huggingface.co/datasets/UKPLab/SLTrans` |
| ProGraML | v0.3.2 | Apache 2.0 | `https://github.com/ChrisCummins/ProGraML` |
| ComPile | v1.0 | CC BY 4.0 | `https://huggingface.co/datasets/llvm-ml/ComPile` |

