# OpenReview forum: "IR-OptSet: An Optimization-Sensitive Dataset for Advancing LLM-Based IR Optimizer"
_NeurIPS.cc/2025/Datasets_and_Benchmarks_Track — NeurIPS 2025 Datasets and Benchmarks Track poster_

### Official Review · Reviewer_mG8v · 2025-07-01

**Rating:** 5
**Confidence:** 4

**Summary:**

The paper introduces IR-OptSet, a new and publicly available dataset designed to enhance the capabilities of large language models in performing IR-level compiler optimizations. Traditional compilers like LLVM rely heavily on manually crafted transformation rules applied to IR, a process that becomes increasingly unscalable. LLMs have been limited by the lack of datasets rich in optimization-sensitive IR examples. IR-OptSet provides a diverse and representative collection of IR samples that expose models to real-world transformation patterns. IR-OptSet contains 170,564 LLVM IR samples extracted from 1,704 open-source repositories spanning 8 diverse domains, including high-performance computing, machine learning, embedded systems, and multimedia. Each sample undergoes an average of 22.89 effective optimization steps, ensuring the dataset captures a wide range of compiler behaviors. IR-OptSet defines two core tasks: Code Analysis, which includes sub-tasks such as dominator tree construction, loop detection, and memory access analysis; and Optimized Code Generation, which requires models to produce optimized IR from its unoptimized counterpart. These tasks are supported by an integrated toolchain for correctness verification (via LLVM and Alive2), static performance evaluation (using llvm-mca), and dataset extensibility.

Experimental results show that fine-tuning both specialized (LLM Compiler) and general-purpose code models (e.g., StarCoder2, Qwen2.5-Coder) on IR-OptSet yields substantial improvements in both understanding and generating optimized IR. The LLM Compiler fine-tuned with IR-OptSet significantly increased its correctness from 6% to over 84%, and in 64 out of 422 test cases, the model generated optimizations that outperformed LLVM’s -O3 optimization level. Comparative evaluations also show that IR-OptSet offers more diverse and effective transformations than existing IR-oriented datasets like SLTrans, ProGraML, and ComPile. The dataset’s diversity and representativeness enable models trained on it to generalize more effectively across unseen samples and alternative datasets.

**Additional Feedback:**

Please see the weaknesses.

**Dataset Code Accessibility:**

Yes

**Dataset Code Comments:**

Both code and datasets are available publicly. The dataset is hosted on HuggingFace in its final usable form, with clear annotations. The GitHub repository includes comprehensive scripts for data preprocessing, model fine-tuning, and evaluation.

**Ethical Considerations:**

No, there are no or only very minor ethics concerns

**Final Justification:**

The paper is well written, and its presentation is easy to follow. This paper contributes a useful dataset for intermediate representation optimization. The open-sourced data and code make this work easy to use and can be built on in the future. The analysis demonstrates the benefits of using this data in code analysis and code generation in language models.

**Limitations Weaknesses:**

- IR-OptSet splits IR samples at the function level to stay within LLM context window constraints. It would be better to discuss further how the dataset can be extended to support cross-function optimization scenarios.

- Why does the paper not report error bars or confidence intervals? Given the small-scale subset (6,000 samples) in some evaluations, it would be better to include statistical measures.

- When the model is fine-tuned on IR datasets, would the model lose some of its other code generation capabilities? It would be better to include a discussion in this direction.

**Strengths Contributions:**

- The introduction of IR-OptSet addresses a crucial gap in the application of large language models to intermediate representation optimization. This is an area previously underexplored. This work offers a novel and high-impact dataset that empowers LLMs to learn more adaptive and generalizable transformation strategies. This dataset includes over 170K IR samples from a wide array of real-world domains.

- The paper compares IR-OptSet with established datasets like SLTrans, ProGraML, and ComPile, and demonstrates its superiority in terms of transformation diversity, representativeness, and downstream performance impact.

- The authors show that models fine-tuned on IR-OptSet not only outperform their counterparts on standard metrics but can also generate optimized IR that exceeds the performance of traditional compiler outputs (e.g., LLVM -O3).

- The paper is well-structured, logically organized, and written in a clear, accessible manner.

---

> ### Author Rebuttal · Authors · 2025-07-31
>
> Thanks for your constructive comments.
>
> ##  Q1.Inclusion of samples in IR-OptSet
>
> As stated in lines 139 – 140 in our paper, the design of IR-OptSet follows prior work [1] by providing only single-function samples. This choice is primarily motivated by two considerations:
>
> - Currently, LLM-based IR optimizers [1][2] focus mainly on single-function analysis and optimization, and there remains substantial room for progress in single-function optimization.
>
> - The tool used to verify functional equivalence of LLM-optimized IR, Alive2, supports automated semantic verification only within single functions, and does not yet handle multi-function scenarios [3]. Consequently, the equivalence of LLM-optimized multi-function IR cannot yet be automatically verified.
>
> Despite the current lack of support for multi-function IR in existing tools, we find your suggestion highly insightful. With future advancements in verification tools such as Alive2, we plan to extend IR-OptSet by incorporating multi-function samples to enable research on multi-function optimization.
>
>
> \[1] Cummins C, Seeker V, Grubisic D, et al. Large language models for compiler optimization. arXiv:2309.07062, 2023.
>
> \[2] Fang X, Mukhanov L. Towards LLM-based optimization compilers. Can LLMs learn how to apply a single peephole optimization? Reasoning is all LLMs need!. arXiv:2412.12163, 2024.
>
> \[3] Lopes N P, Lee J, Hur C K, et al. Alive2: bounded translation validation for LLVM[C]//Proceedings of the 42nd ACM SIGPLAN International Conference on Programming Language Design and Implementation. 2021: 65-79.
>
>
> ## Q2. Confidence Intervals
>
> Due to time constraints, we focused on the most challenging task -- Optimized Code Generation -- and conducted three independent runs to assess result consistency and calculate the confidence interval. The results, summarized in the table below, demonstrate stable performance across runs. Reported values represent the mean ± 95% confidence interval. **We will include this information in the revised version**.
>
>
> | Model         | EM(%)        | BLEU        | Corr(%)      |
> | ------------- | ------------ | ----------- | ------------ |
> | LLM Compiler  | 53.80 ± 4.07 | 0.96 ± 0.04 | 85.33 ± 2.24 |
> | StarCoder2    | 4.60 ± 1.31  | 0.68 ± 0.05 | 56.20 ± 4.74 |
> | Qwen2.5-Coder | 3.00 ± 1.79  | 0.79 ± 0.02 | 44.00 ± 2.17 |
>
> ## Q3.Potential impact on other code generation capabilities
>
> Our dataset is primarily designed to help LLMs better comprehend and perform IR-level optimizations. While it is not intended for general-purpose code generation, prior work such as IRCoder \[1] and TransCoder-IR \[2] has demonstrated that incorporating IR into training dataset can improve prompt robustness, multilingual code completion, code understanding, and instruction following. Therefore, introducing IR-level optimization tasks may not only enhance optimization performance but also contribute to more robust and generalizable code understanding.
>
> \[1] Paul I, Glavaš G, Gurevych I. IRCoder: Intermediate Representations Make Language Models Robust Multilingual Code Generators. (ACL 2024)
>
> \[2] Szafraniec M, Roziere B, Leather H, et al. Code translation with compiler representations. arXiv:2207.03578, 2022.

---

### Official Review · Reviewer_gELF · 2025-07-02

**Rating:** 4
**Confidence:** 2

**Summary:**

This paper presents IR-OptSet, the first large-scale (170K samples) optimization-sensitive dataset for LLM-based compiler optimization, addressing the critical lack of diverse IR transformation samples needed for effective model training. The dataset spans 8 optimization domains and introduces two core tasks (Code Analysis and Optimized Code Generation) along with verification and evaluation tools, enabling systematic benchmarking. Experiments demonstrate that LLMs fine-tuned on IR-OptSet achieve significant performance gains, even surpassing traditional O3 compiler optimizations in 64 test cases, while analysis shows superior diversity compared to existing datasets.

**Dataset Code Accessibility:**

Yes

**Ethical Considerations:**

No, there are no or only very minor ethics concerns

**Final Justification:**

Thank you for the clarifications provided in the rebuttal. As I am not a domain expert in this field, I will maintain my original rating.

**Limitations Weaknesses:**

The reviewer questions whether this paper's topic falls within NeurIPS's scope or might be better suited for architecture-focused conferences. The reviewer recommends that the authors provide more illustrative examples to better explain the task and make it more accessible to readers.

**Strengths Contributions:**

- The paper introduces IR-OptSet, the first public optimization-sensitive dataset for LLM-based compiler optimization, addressing a critical gap with 170K diverse LLVM IR samples across 8 optimization domains.

- It defines two core tasks (Code Analysis and Optimized Code Generation) and provides verification/evaluation tools, enabling systematic benchmarking and future dataset expansion.

- Experiments show that LLMs fine-tuned on IR-OptSet achieve significant accuracy gains and even outperform traditional O3 optimizers in 64 test cases, demonstrating practical utility.

---

> ### Author Rebuttal · Authors · 2025-07-31
>
> Thanks for your constructive comments.
>
> ## Q1.Scope justification
>
> With the rapid advancement of LLMs, there has been a growing trend toward applying LLMs to computer systems and architecture design at top-tier AI venues \[1-5]. Our work follows this direction by advancing LLM-based IR optimizers in compilers, contributing to this emerging intersection between AI and computer systems.
>
> \[1\] Zhong M, Lyu F, Wang L, et al. Comback: A versatile dataset for enhancing compiler backend development efficiency[J]. Advances in Neural Information Processing Systems, 2024, 37: 112310-112328. (NeurIPS 2024)
>
> \[2\] Zheng L, Liu R, Shao J, et al. Tenset: A large-scale program performance dataset for learned tensor compilers[C]//Thirty-fifth Conference on Neural Information Processing Systems Datasets and Benchmarks Track (Round 1). 2021. (NeurIPS 2021)
>
> \[3\] Jiang X, Zhao Y, Lin Y, et al. Circuitnet 2.0: An advanced dataset for promoting machine learning innovations in realistic chip design environment[C]//The Twelfth International Conference on Learning Representations. 2024. (ICLR 2024)
>
> \[4\] Chen H, Wen Y, Cheng L, et al. Autoos: make your os more powerful by exploiting large language models[C]//Forty-first International Conference on Machine Learning. 2024. (ICML 2024)
>
> \[5\] Cheng S, Jin P, Guo Q, et al. Automated CPU design by learning from input-output examples[C]//Proceedings of the Thirty-Third International Joint Conference on Artificial Intelligence. 2024: 3843-3853. (IJCAI 2024)
>
>
> ## Q2.Clarifying the Task with Illustrative Examples
>
> We have provided an example in Figure 3 of our paper and illustrated our tasks in Section 3.3. To further enhance clarity, we will add step-by-step explanations in the appendix to show how the ground truth in Figure 3 is derived from the input, along with additional representative examples to aid understanding.

---

### Official Review · Reviewer_U7G6 · 2025-07-02

**Rating:** 4
**Confidence:** 3

**Summary:**

The paper introduces IR-OptSet,  a publicly available optimization-sensitive dataset for LLM IR optimizers, addressing the challenges of scaling rule-based compiler optimization. The dataset provides two core tasks along with three tools for robust evaluation. Experiments show that LLMs fine-tuned on IR-OptSet improve their accuracy of code analysis and optimized code generation, surpassing a traditional compiler on some test cases. Also, the authors use LLM Compiler to demonstrate the transformation diversity and representativeness of IR-OptSet. The paper provides extensive technical details for their project.

**Additional Feedback:**

See weaknesses for questions.

**Dataset Code Accessibility:**

Yes

**Dataset Code Comments:**

The dataset is publicly accessible through HuggingFace. The code/scripts for reproduction are provided via GitHub.

**Ethical Comments:**

The author claims that IR-OptSet does not contain any personally identifiable information or offensive content.

**Ethical Considerations:**

No, there are no or only very minor ethics concerns

**Final Justification:**

Overall, I recommend accepting this work. The primary contribution, providing an optimization-sensitive dataset for LLM IR optimizers, is fairly solid. My main concern lies with the clarity of the manuscript, as some technical details of implementing the synthesis and training are obscure, and some experimental settings are not robust enough. The revised version successfully addresses the majority of these concerns and demonstrates sufficient quality to warrant acceptance.

**Limitations Weaknesses:**

1. The organization of the paper should be improved. Why are the RQs in experiments important, provided they are answered by experiments? It should be clearly stated in the introduction.
2. Comparison of the capacity of different datasets is missing.
3. Some experiment choices are not explained, e.g., why use a small subset for fine-tuning, rather than all data within the context window?
4. The chapter of experiments is messy and hard to interpret. e.g., line 305, what does this example imply? It seems unrelated to the context. Also, some key hyperparameters, including LoRa and should be included in the body.
5. The choice of data splits seems unreasonable. Train, validation, and test sets are randomly selected from the same distribution. This makes the comparison with other datasets and models without fine-tuning unfair. Honestly, I cannot infer whether this data split is the same as RQ 3.
6. Why is Opt. Eff. defined but not shown in experiments? The author claims the dataset to be optimization-sensitive, but an important metric is missing.
7. The experiment uses a small-sized version for Qwen2.5-Coder, which is an unnatural choice because clearly the authors have the resources to run 7B fine-tunes. I think larger versions should be included.
8. RQ.2 is a bit overclaimed, given that 51 cases are worse than LLVM-O3. The significance of 64 better cases should be properly assessed.

**Strengths Contributions:**

1. The motivation of the paper is well illustrated and explained.
2. The source of LLVM IR is comprehensive.

---

> ### Author Rebuttal · Authors · 2025-07-31
>
> Thanks for your constructive comments.
>
> ## Q1.Organization of the RQs
>
> Our experiments focus on IR-OptSet as an optimization-sensitive dataset for advancing LLM-based IR optimization. RQ1 and RQ2 investigate whether IR-OptSet enables LLMs to better comprehend and perform IR-level optimizations. RQ3 compares IR-OptSet with existing datasets to assess whether it offers a more diverse and representative set of optimization-sensitive transformations that further support LLM-driven compiler optimization. The key challenges and insights related to these research questions are discussed in lines 44 - 53 in the Introduction of our paper.
>
> Regarding the potential overstatement in RQ2, we agree that the original claim may be too strong and will revise it to more cautiously state that **LLMs have the potential to outperform traditional compilers**.
>
>
> ## Q2.Comparison of different datasets
>
> In the revised version, we will include the following comparison table that highlights the differences between IR-OptSet and other IR-oriented datasets in terms of the number of samples, source repositories, dataset objectives, available toolchains, and the Average Effective Optimization Step (a metric we introduce to quantify optimization sensitivity).
>
> | Dataset              | Samples | Source Repos | Dataset Objective                        | Toolchain                                                   | Avg. Eff. Opt. Steps |
> | -------------------- | -------- | ------------- | ---------------------------------------- | ----------------------------------------------------------- | -------------------- |
> | **IR-OptSet (Ours)** | 170K     | 1704          | Code Analysis, Optimized Code Generation | Correctness Verification, Performance Evaluation, Extension | 25.50                |
> | SLTrans              | 6.9M     | -             | Neural Code Translation                  | -                                                           | 21.92                |
> | ProGraML             | 469K     | -             | Code Analysis                            | -                                                           | 13.33                |
> | ComPile              | 1.9T    | -             | Code Analysis, Optimized Code Generation | Extension                                                   | 10.60                |
>
> Although IR-OptSet contains fewer samples, it achieves a substantially **higher Average Effective Optimization Step**, indicating richer optimization behavior per sample, underscoring IR-OptSet's value as an optimization-sensitive dataset.
>
>
> ## Q3.Experiment settings
>
> * Regarding the use of a small subset for fine-tuning, as noted in lines 270 – 271, we adopt this setting to demonstrate that even limited-scale training on IR-OptSet enables LLMs to better comprehend and perform IR-level optimizations, underscoring the quality of IR-OptSet. To support the robustness of our findings under this setup, we report 95% confidence intervals (CIs) based on three independent runs for the most challenging task (Optimized Code Generation) in IR-OptSet. This single task was selected due to time constraints during the author response period. Values in the table below represent the mean ± 95% CI. These results confirm the stability and effectiveness of IR-OptSet, even under small-scale fine-tuning.
>
>
>     | Model         | EM(%)        | BLEU        | Corr(%)      |
>     | ------------- | ------------ | ----------- | ------------ |
>     | LLM Compiler  | 53.80 ± 4.07 | 0.96 ± 0.04 | 85.33 ± 2.24 |
>     | StarCoder2    | 4.60 ± 1.31  | 0.68 ± 0.05 | 56.20 ± 4.74 |
>     | Qwen2.5-Coder | 3.00 ± 1.79  | 0.79 ± 0.02 | 44.00 ± 2.17 |
>
>
>
> * In response to concerns about the fairness of using a small subset to compare IR-OptSet with other IR-oriented datasets, we conducted a larger-scale experiment on the Optimized Code Generation task, fine-tuning the LLM Compiler on 10,000 samples from each dataset. This resulted in four variants: LLM Compiler FTD IR-OptSet, SLTrans, ProGraML, and ComPile. The test set follows the same setup described in lines 296 – 299, consisting of 1,000 samples with 250 randomly selected from each dataset. The results below confirm the same trend observed in the smaller-scale setting (5000), further validating the effectiveness of IR-OptSet.
>
>     | LLM Compiler      | EM (5000)(%) | EM (10000)(%) | BLEU (5000) | BLEU (10000) | Corr. (5000)(%) | Corr. (10000)(%) |
>     | ----------------- | ------------ | ------------- | ----------- | ------------ | --------------- | ---------------- |
>     | **FTD IR-OptSet** | 45.40        | 44.00         | 0.86        | 0.87         | 81.00           | 81.20            |
>     | FTD  SLTrans      | 40.60        | 36.40         | 0.86        | 0.87         | 75.40           | 77.80            |
>     | FTD  ProGraML     | 27.00        | 34.20         | 0.66        | 0.84         | 70.60           | 79.00            |
>     | FTD ComPile       | 43.40        | 35.40         | 0.85        | 0.88         | 76.60           | 76.40            |
>
>
> * For the concern regarding data split and fairness of comparison, as detailed in lines 296 – 299, we fine-tuned the LLM Compiler on 5,000 samples from each dataset, resulting in four variants: LLM Compiler FTD IR-OptSet, SLTrans, ProGraML, and ComPile. The test set comprises 500 samples in total, with 125 randomly selected from each dataset, ensuring a balanced and fair evaluation under equal data volume.
>
>
>
>
> * For the comparison on optimization sensitivity, we provide a detailed analysis in lines 288 – 295. Specifically, IR-OptSet consistently yields the highest effectiveness frequency across nearly all of the top-30 optimization steps when compared to other datasets, reinforcing its role as an optimization-sensitive dataset.
>
> ## Q4. Writings
>
> * Regarding the clarity of the experimental section, the example in line 305 illustrates the results in Figure 6. It shows that each fine-tuned model performs best on samples from its own training dataset (e.g., LLM Compiler fine-tuned on ComPile achieves the highest accuracy on ComPile samples). However, the LLM Compiler fine-tuned on IR-OptSet consistently outperforms all other fine-tuned models on datasets outside their own (e.g., LLM Compiler fine-tuned on IR-OptSet achieves the second-highest accuracy on ComPile samples), demonstrating stronger cross-dataset generalization ability. We will clarify this explanation in the revised version.
>
> * For the missing hyper parameters, we have included them in the appendix due to space constraints and will move them to the main text in the revision.
>
> * For the absence of Optimization Effectiveness (Opt. Eff.) in the paper, we clarify that it is in fact presented on the y-axis of Figure 4. However, the y-axis was mislabeled as "Perf. Eff." We will correct this y-axis label to "Opt. Eff. vs LLVM -O3" in the revised version.
>
>
> ## Q5. Qwen2.5-Coder-7B
>
> Our experiments span models of varying sizes, from LLM Compiler (7B) to Qwen2.5-Coder (1.5B). Since LLM Compiler already represents a 7B-scale model, we did not initially include Qwen2.5-Coder-7B in our main experiments. To further explore the effect of IR-OptSet on Qwen2.5-Coder-7B, we conducted follow-up experiments, which show that it also benefits from fine-tuning on IR-OptSet. Due to time constraints, we focused on the most challenging task (Optimized Code Generation) to illustrate this effect.
>
> |                    | EM(%) | BLEU | Corr(%) |
> | ------------------ | ----- | ---- | ------- |
> | Qwen2.5-Coder-1.5B | 2.20  | 0.79 | 43.60   |
> | Qwen2.5-Coder-7B   | 6.00  | 0.93 | 52.80   |

---

> > ### Comment · Reviewer_U7G6 · 2025-08-08
> >
> > Thanks for the response. I believe my concerns are well addressed, and I have decided to raise my rating.

---

> ### Author Response · Authors · 2025-08-06
>
> Dear Reviewer,
>
> I hope this message finds you well. Thank you for the time and effort you've dedicated to reviewing our paper. We've carefully addressed your comments in our rebuttal and would greatly appreciate any further feedback. If there are any remaining concerns, we'd be happy to clarify.
>
> Best regards,
>
> The Authors

---

### Official Review · Reviewer_ivan · 2025-07-19

**Rating:** 5
**Confidence:** 2

**Summary:**

In this submission, the authors start by stressing that traditional compilers rely on manually crafted transformation rules apply to IR. As their complexity increase they become difficult to maintain and extend. But Large Language Models (LLM) can offer an alternative, however they need IR-oriented datasets to generate appropriate IR transformations.

This is why the authors are introducing a dataset of optimized IR samples, called IR-OptSet, for advancing LLM-based IR optimizers. The datasets includes 170K LLVM IR samples collected from 1704 OSS repositories and over 8 optimization domains. This dataset enables two different type of tasks which are simply modelled against the compiler optimization pipeline:
- code analysis with three sub-tasks: dominator tree constructions, loop detection and memory access analysis, which are amenable to automated data generation as LLVM uses a human-readable format for their results, and
- optimized code generation measured by the analysis of cycles required to execute the generated IR translated to assembly.

The authors also provide (and document) tooling associated to this dataset for verification, evaluation and expansion.

Thanks to this new dataset, the authors are able to demonstrate a number of results:
- Finetuning an LLM (the authors picked LLM Compiler FTD 7B, StarCoder2-3B and Qwen2.5-Coder-1.5B) on (a subset of) IR-OptSet substantially improves performance on code analysis and optimization code generation (Table 2),
- An LLM finetuned on IR-OptSet can outperform LLVM IR -O3 in 64 test cases.

Additionally the authors illustrate how IR-OptSet help LLM generalize better compare to alternative IR datasets (ComPile, ProGraML and SLTrans).

**Dataset Code Accessibility:**

Yes

**Dataset Code Comments:**

The full dataset is available on HuggingFace at the URL shared by the authors. I did not download it, but did browse and inspect several samples. Each sample IR is included in raw, preprocessed and optimized forms with metadata like active optimization passes or source repository, file path and function name (as the samples are only function-level).

In terms of code, the authors also provide on their GitHub repository (as promised on the paper), a small toolchain to prepare, validate and evaluate IRs. The paper and repository include instructions on how to use these tool for reproduction.

Overall, this definitely looks like a quality dataset preparation ready for release and direct use by the community.

**Ethical Considerations:**

No, there are no or only very minor ethics concerns

**Final Justification:**

First, I would like to thank the authors for their rebuttal and answers to each reviewer. I have read these answers and followed the discussion and I found my concerns or remaining questions well addressed. I am thus keeping my recommendation to Accept: the paper tackles an important problem for the community and the execution (dataset preparation and experimental validation) is solid.

**Limitations Weaknesses:**

- IR tend to be quite verbose and can easily lead to a greater number of tokens than the LLM context window. While the authors acknowledge this and point to LTO so solve cross-function optimization, it would have been beneficial to include samples with multiple functions (and associated calls) that either could still possibly fit within the largest context window of LLMs or may just be available in the dataset for future reference as approaches eventually manage to handle them.

- While the paper correctly summarizes high-level findings and does a good job at selling the value or IR-OptSet, more in-depth analysis would further improve it. In particular:
  - For correctness: are there any observable failure patterns that make LLMs trip?
  - For optimization: it appears that finetuning on IR-OptSet can help LLMs outperform -O3 level of optimization. Given this achievement and the number of samples (64) where this was observed, is there similarly a pattern where LLMs shine compared to established compilers?

- The related work section is covering too lightly IR-oriented datasets. A discussion and a comparison table with e.g. number of samples, repository, optimization or not applied would be needed to better position the contributions of the authors.

- More details on the dataset construction could also be included (Section 3.2) and be very informative and valuable for the community: how did the authors end up with 170K samples from 1.7K repositories. What steps in the automated pipeline tend to fail the most, what prevented them from extracting more data from GitHub, etc...

**Strengths Contributions:**

- The paper is well written and its presentation made it easy to follow and understand the specific contributions of the authors.

- The authors do a good job at demonstrating the value of their new dataset: improving performance of LLM of the two target tasks, better generalization than other available IR datasets, even (in some cases) outperforming regular optimizers.

- The dataset preparation and code release (see below in dataset accessibility for details) is solid. In particular the provided toolchain can help future-proof this contribution and allow the community to expand it.

- The high-level problem tackled by the authors (improving ML-based optimizers and moving away from manual optimization rules) is quite relevant to a significant part of the community working on LLM for code. Consequently the proposed dataset should easily find users and foster research in this area.

---

> ### Author Rebuttal · Authors · 2025-07-31
>
> Thanks for your constructive comments.
>
> ##  Q1.Inclusion of multi-function samples in IR-OptSet
>
> As stated in lines 139 – 140 in our paper, the design of IR-OptSet follows prior work [1] by providing only single-function samples. This choice is primarily motivated by two considerations:
>
> - Currently, LLM-based IR optimizers [1][2] focus mainly on single-function analysis and optimization, and there remains substantial room for progress in single-function optimization.
>
> - The tool used to verify functional equivalence of LLM-optimized IR, Alive2, supports automated semantic verification only within single functions, and does not yet handle multi-function scenarios [3]. Consequently, the equivalence of LLM-optimized multi-function IR cannot yet be automatically verified.
>
> Despite the current lack of support for multi-function in existing tools, we find your suggestion highly insightful. With future advancements in verification tools such as Alive2, we plan to extend IR-OptSet by incorporating multi-function samples to enable research on multi-function optimization.
>
>
>
> \[1] Cummins C, Seeker V, Grubisic D, et al. Large language models for compiler optimization. arXiv:2309.07062, 2023.
>
> \[2] Fang X, Mukhanov L. Towards LLM-based optimization compilers. Can LLMs learn how to apply a single peephole optimization? Reasoning is all LLMs need!. arXiv:2412.12163, 2024.
>
> \[3] Lopes N P, Lee J, Hur C K, et al. Alive2: bounded translation validation for LLVM[C]//Proceedings of the 42nd ACM SIGPLAN International Conference on Programming Language Design and Implementation. 2021: 65-79.
>
> ## Q2. In-depth analysis
>
> * **For correctness**: we conducted an in-depth analysis of the 78 failed cases made by the LLM Compiler in the Optimized Code Generation task (Table 2 in our paper), and identified three common error patterns. These errors mainly stem from the model's inaccurate reasoning about data flow  --  that is, how values are defined, propagated, and used across different branches of the program.
>
>   * Define-use error (21 cases):
>
>     Example: Use of undefined value `%7`
>
>     ```
>     %6 = add i32 %4, %5
>     %8 = mul i32 %6, %7
>     ```
>
>   * Structural violations in PHI nodes (18 cases):
>
>     Example: %B3 is not a predecessor of %B2
>
>     ```
>     ; B2 has two predecessors: B1 and B4
>     B2:
>     %5 = phi i32 [ 0, %B1 ], [ %10, %B3 ]
>     ```
>
>   * Data type error (7 cases):
>
>     Example: `%10` defined as `i32`, but used as `double`
>
>     ```
>     %10 = add i32 %0, 1
>     %11 = fcmp ult double %8, %10
>     ```
>
>
> * **For performance**: We further analyzed the 64 cases in Figure 4 where the LLM outperforms the traditional compiler and categorized them into three common patterns. Note that some cases fall into multiple categories, collectively contributing to their performance gains over LLVM; as a result, the total number of cases in the table below exceeds 64. These patterns highlight scenarios in which the LLM exhibits stronger optimization capabilities.
>
>   * Rewriting complex conditions into simple ones (32 cases):
>
>     LLVM:
>
>     ```
>     switch i32 %0, label %B1 [
>      i32 0, label %B1
>      i32 1, label %B2
>     ]
>     ```
>
>     LLM Compiler FTD IR-OptSet:
>
>     ```
>     %1 = icmp eq i32 %0, 1
>     br i1 %1, label %B2, label %B1
>     ```
>
>   * Removing unnecessary computations (24 cases):
>
>     LLVM:
>
>     ```
>     %84 = add nsw i64 %i, 1
>     %85 = add nsw i64 %84, 2
>     ```
>
>     LLM Compiler FTD IR-OptSet:
>
>     ```
>     %85 = add nsw i64 %i, 3
>     ```
>
>   * Replacing verbose code with built-in functions
>
>     LLVM (a loop that writes zero values to an array):
>
>     ```
>     loop:
>     %i = phi i64 [ 0, %entry ], [ %next, %loop ]
>     %ptr = getelementptr <2 x float>, ptr %p, i64 %i
>     store <2 x float> 0.0, ptr %p
>     %next = add i64 %i, 2
>     %cond = icmp ult i64 %next, %size
>     br i1 %cond, label %loop, label %exit
>     ```
>
>     LLM Compiler FTD IR-OptSet:
>
>     ```
>     call void @llvm.memset.p0.i64(ptr %p, i8 0, i64 %size)
>     ```
>
> We will add these details in the revision.
>
> ## Q3.Related work
>
> Thanks for the suggestion. In the revision, we will add **the following comparison table** in the Related Work section, covering the number of samples, source repositories, dataset objectives, provided toolchains, and the Average Effective Optimization Step -- a metric we introduce to quantify optimization sensitivity. A dash ("–") is used where data is not publicly available.
>
> Although IR-OptSet contains fewer samples, it achieves a substantially **higher Average Effective Optimization Step**, indicating richer optimization behavior per sample and underscoring IR-OptSet's value as an optimization-sensitive dataset.
>
>
> | Dataset              | Samples | Source Repos | Dataset Objective                        | Toolchain                                                   | Avg. Eff. Opt. Steps |
> | -------------------- | ------- | ------------ | ---------------------------------------- | ----------------------------------------------------------- | -------------------- |
> | **IR-OptSet (Ours)** | 170K    | 1704         | Code Analysis, Optimized Code Generation | Correctness Verification, Performance Evaluation, Extension | 25.50                |
> | SLTrans              | 6.9M    | -            | Neural Code Translation                  | -                                                           | 21.92                |
> | ProGraML             | 469K    | -            | Code Analysis                            | -                                                           | 13.33                |
> | ComPile              | 1.9T    | -            | Code Analysis, Optimized Code Generation | Extension                                                   | 10.60                |
>
> ## Q4.Details on IR-OptSet construction
>
> The construction process followed an automated pipeline with the following steps:
>
> 1. **Code Collection**: We crawled GitHub using the keywords detailed in Appendix A, identifying 4,031 relevant repositories. We filtered out empty repositories and those with licensing issues, resulting in 1,704 repositories.
> 2. **LLVM IR Extraction**: Using `clang -Xclang -disable-llvm-passes -emit-llvm`, we extracted LLVM IR directly after the raw code was processed by the LLVM 19.1.0 frontend, before any optimizations. We filtered out files dependent on specific operating systems, external libraries, or incomplete projects, and utilized `llvm-extract` and StructuralHash, yielding 260K non-redundant LLVM IR files.
> 3. **IR Normalization**: We applied an automated script to standardize the IR files.
> 4. **Optimization Annotation**: We filtered out IR files where fewer than 8% of the executed passes resulted in transformations, classifying them as optimization-insensitive. This process resulted in 170K finalized LLVM IR files.
>
> In terms of failures in the pipeline, we found that filtering out files dependent on specific operating systems, external libraries, and incomplete projects were the most challenging steps.

---

> > ### Comment · Reviewer_ivan · 2025-08-06
> >
> > Thanks for your thorough reply, I believe this addressed most of the concerns or questions I still had.

---

### Decision · Program_Chairs · 2025-09-18

**Decision:**

Accept (poster)

**Comment:**

This paper presents IR-OptSet, a novel dataset containing 170K LLVM IR samples specifically designed for training LLMs on compiler optimization tasks. The work addresses a critical gap in available datasets for LLM-based compiler optimization by providing optimization-sensitive samples that expose models to diverse transformation patterns in real-world scenarios.

Technical Merit and Contributions:
Strong Dataset Contribution: IR-OptSet represents the first public dataset specifically focused on optimization-sensitive IR samples. The authors curated data from 1,704 GitHub repositories across 8 optimization domains, with each sample undergoing an average of 22.89 effective optimization steps. This focus on optimization sensitivity distinguishes it from existing IR datasets that may lack transformative diversity.

Comprehensive Evaluation Framework: The paper defines two well-motivated tasks (Code Analysis and Optimized Code Generation) aligned with compiler optimization workflows, supported by verification tools for correctness and performance evaluation. The experimental evaluation spans multiple model architectures and provides thorough baselines.

Practical Impact: Fine-tuning experiments demonstrate significant improvements across all tested models, with the LLM Compiler achieving substantial gains (EM: +52.00%, Correctness: +78.40% for Optimized Code Generation). Notably, the fine-tuned model outperformed LLVM -O3 in 64 out of 422 test cases, demonstrating practical potential.

Areas of Concern and Limitations:
Limited Performance Against Traditional Compilers: the concern "average performance compared to O3" is very valid and and represents a key limitation. While the paper claims LLMs outperformed LLVM -O3 in 64 cases, this must be contextualized:
- The model performed worse than -O3 in 51 cases and equal in 307 cases
- This means superior performance in only ~15% of cases (64/422)
- The paper's framing could be seen as overstating the practical advantages

However, this limitation could be viewed in the context of:
- Early-stage research in LLM-based compilation
- Proof-of-concept that LLMs can sometimes exceed traditional optimizers
- Foundation for future improvements rather than production-ready replacement

Missing Statistical Analysis: Multiple reviewers noted the absence of error bars and confidence intervals. While authors provided some confidence intervals in rebuttals for the most challenging task, the main results lack proper statistical validation.
Scope Limitations: The dataset focuses on single-function optimization due to current tool limitations (Alive2 verification), which restricts cross-function optimization research. The authors acknowledge this but provide reasonable justification based on current tooling constraints.
Context Window Constraints: The 4K token limit necessitates function-level splitting, though authors provide reasonable mitigation strategies through Link Time Optimization approaches.

Reviewer Assessment:
All four reviewers ultimately recommended acceptance (ratings 4-5), recognizing the solid technical contribution despite presentation and evaluation concerns. Reviewers appreciated the comprehensive dataset construction, thorough experimental evaluation, and potential community impact.